# In Search of Adam's Secret Sauce

**Antonio Orvieto** *
ELLIS Institute Tübingen, MPI-IS
Tübingen AI Center, Germany

**Robert M. Gower**
CCM, Flatiron Institute, Simons Foundation
New York, US

## Abstract

Understanding the remarkable efficacy of Adam when training transformer-based language models has become a central research topic within the optimization community. To gain deeper insights, several simplifications of Adam have been proposed, such as the signed gradient and signed momentum methods. In this work, we conduct an extensive empirical study — training over 1,500 language models across different data configurations and scales — comparing Adam to several known simplified variants. We find that signed momentum methods are faster than SGD, but consistently underperform relative to Adam, even after careful tuning of momentum, clipping setting and learning rates. However, our analysis reveals a compelling option that preserves near-optimal performance while allowing for new insightful reformulations: constraining the Adam momentum parameters to be equal, $\beta_1 = \beta_2$. Beyond robust performance, this choice affords new theoretical insights, highlights the *"secret sauce"* on top of signed momentum, and grants a precise statistical interpretation: we show that Adam in this setting implements a natural online algorithm for estimating the mean and variance of gradients—one that arises from a mean-field Gaussian variational inference perspective.

## 1 Introduction

Despite a decade of research into efficient and performant adaptive optimizers for deep learning, the *de facto* choice for large-scale training today remains `Adam` [Kingma and Ba, 2014], especially for training language models (LMs) [Grattafiori et al., 2024, Liu et al., 2024]. At the root of this choice is the peculiar geometry of optimization landscapes induced by the transformer architecture [Noci et al., 2022, Zhang et al., 2024a], as well as the noisy/unbalanced nature of tokenized text data [Zhang et al., 2020a, Kunstner et al., 2024].

In recent years, the surge of extremely large-scale and expensive-to-pretrain language models has further pushed the community to better understand `Adam`'s performance and to propose faster, efficient, and robust alternatives. Towards achieving this goal, contemporary studies [Kunstner et al., 2023, Bernstein and Newhouse, 2024] have brought up a close similarity between the performance of `Adam` and `SignSGD` [Bernstein et al., 2018] with momentum. While such results are extremely valuable to forward our understanding, they are not precise enough : already at a scale of 160M parameters we found that extensive tuning of `Signum` (`SignSGD` with momentum), while closing 96% of the perplexity gap between `SGD` and `Adam`, results in a 25% effective slowdown (Figure 1).

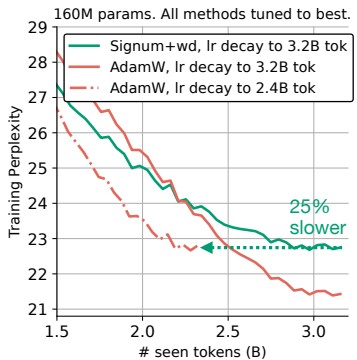

Figure 1: *Pretraining on SlimPajama with Chinchilla-optimal [Hoffmann et al., 2022] scaling. Both momentum and learning rates for `Signum` are extensively tuned (§3). While `Signum` closes* 96% *of the perplexity gap between `Adam` and `SGD` with momentum (Table 1), still results in a* 25% *slowdown : `Adam` achieves the same performance with 3/4 of the budget.*

---

\* antonio@tue.ellis.eu

39th Conference on Neural Information Processing Systems (NeurIPS 2025).

Table 1: ***(Signum closes 96% of the perplexity gap between Adam and SGD)*** *Validation perplexity comparison of widely used optimizers that interpolate between* `SGD` *and* `Adam`*, evaluated on a language modeling task (160M parameters, 3.2B SlimPajama tokens, sequence length 2048, batch size 256 – Chinchilla optimal). We report the mean and 2-sigma interval of validation perplexity (on 100M held-out tokens) across 3 initialization seeds. Weight decay is always decoupled [Loshchilov and Hutter, 2019] and set to* 0.1 *[Biderman et al., 2023, Liu et al., 2024] except for* `SGD` *where we further tune (§B).* `RMSprop` *does not use momentum, and Gclip is global norm clipping to* 1 *(before applying momentum), Cclip is coordinate-wise clipping (after applying momentum). Other hyperparameters, for all other methods, are carefully tuned, see e.g. Figure 2 and §3.*
*To optimally tune hyperparameters (e.g. Figure 2), we performed a total of 582 full training runs.*

|  | Adam | Signum | RMSprop | SGD+Cclip | SignSGD | SGD+Gclip | SGD |
|---|---|---|---|---|---|---|---|
| Val ppl. | **21.86**$\pm$ **0.21** | 23.23$\pm$ 0.16 | 27.04$\pm$ 0.34 | 33.40$\pm$ 0.39 | 36.78$\pm$ 0.57 | 37.76$\pm$ 0.61 | 53.62$\pm$ 5.14 |

While for large-scale training, the slowdown in Figure 1 is not acceptable, it may seem unnecessary or anachronistic to further explain it, in light of recent algorithms claiming to have further improved the performance of `Adam`, e.g. `Muon` [Jordan et al., 2024, Liu et al., 2025, Shah et al., 2025], `Scion` [Pethick et al., 2025], and `Shampoo`-based [Gupta et al., 2018] methods such as `SOAP` [Vyas et al., 2025]. However, a close inspection of such optimizers reveals that, while gains over vanilla `Adam` are solid, *most of these methods still use* `Adam` *on a specific subset of parameters*: For instance, in recent scaled-up versions of `Muon` [Liu et al., 2025, Shah et al., 2025], `Adam` is used to update embedding, LM heads and normalization parameters [2], and on the other parameters the `Muon` update is normalized to have a similar RMS value similar to the `Adam` update. Further, `SOAP`'s improvements stem from the application of `Adam` in the preconditioner's eigenbasis.

The discussion above and the results in Figure 1 inspires us to further dissect – once again [Balles and Hennig, 2018] – the mechanisms of `Adam` compared to those of simpler methods in language modeling with transformers.
Towards improving our understanding of `Adam`, we make the following contributions:

- We perform a large-scale evaluation ($\sim$ 10 thousand NVIDIA A100-SXM4-80GB GPU hours) of the performance of established algorithms which claim a theoretical or empirical similarity/dissimilarity with `Adam` on 160M parameters LMs with usual configurations [Biderman et al., 2023, Black et al., 2022], at a compute-optimal budget on different datasets, at different batch-sizes and sequence lengths (up to 2048 tokens). Crucially, we sweep over all momentum parameters for each method, for each learning rate in our grid – for each of our settings. We find that, while clipping and sign descent methods can close most of the gap with `SGD`, their performance is not satisfactory in comparison to `Adam` (Figure 2). We make all of our data, e.g. loss dynamics for all our settings, publicly available at https://github.com/aorvieto/SecretSauce.

- Through our extensive tuning of `Adam` (e.g., Figure 2, comprising 200 distinct hyperparameter settings), we identify one simplification that does perform well: that of setting $\beta_1 = \beta_2$ (emerging practical choice in contemporary literature [Zhao et al., 2025, Shah et al., 2025, Cattaneo and Shigida, 2025, Zhang et al., 2025]). We validate this finding (§3.2) at different batchsizes, data source, token budget, sequence length and larger scale (410M): $\beta_1 = \beta_2$ performs at near-optimality across the majority of our experiments, see Figure 3. Given the breadth of our evaluation and the robustness of this finding, we recommend adopting $\beta_1 = \beta_2$ as the default setting for Adam for training language models at similar data and parameter scales. More broadly, this perspective suggests that Adam can be effectively treated as a one-parameter optimizer (as `Signum`).

- We show in §4, that reducing $\beta_1 = \beta_2 = \beta$ to a single parameter, leads to a surprising new interpretation of `Adam`: it is built on top of a nontrivial yet principled online method for estimating mean and variance of the gradients. Indeed, we can view the two momentum buffers as the result of an online Gaussian Variational inference method for tracking the mean and variance of the gradients as they change across iterations. This viewpoint directly adds to the discussion by Balles and Hennig [2018], yet affords more precision induced by our empirically-informed simplification.

- We offer a toy quadratic example illustrative of our findings, building on top of recent works on the peculiar landscape of transformer-based language modeling problems [Noci et al., 2022, Zhang et al., 2024a]. This example replicates the gaps between tuned SGD, `Signum`, and `Adam` with equal betas in a 9-dimensional setting, helpful for future research and to gain intuition.

---

[2]Coincidentally, the ones that were shown to be most sensitive during training [Zhao et al., 2025, Kunstner et al., 2024]. `Scion` claims a greater independence from `Adam`, yet adopts an architecture where normalization layers have no trainable gain parameters. While results are promising, experiments in the usual setup are needed.

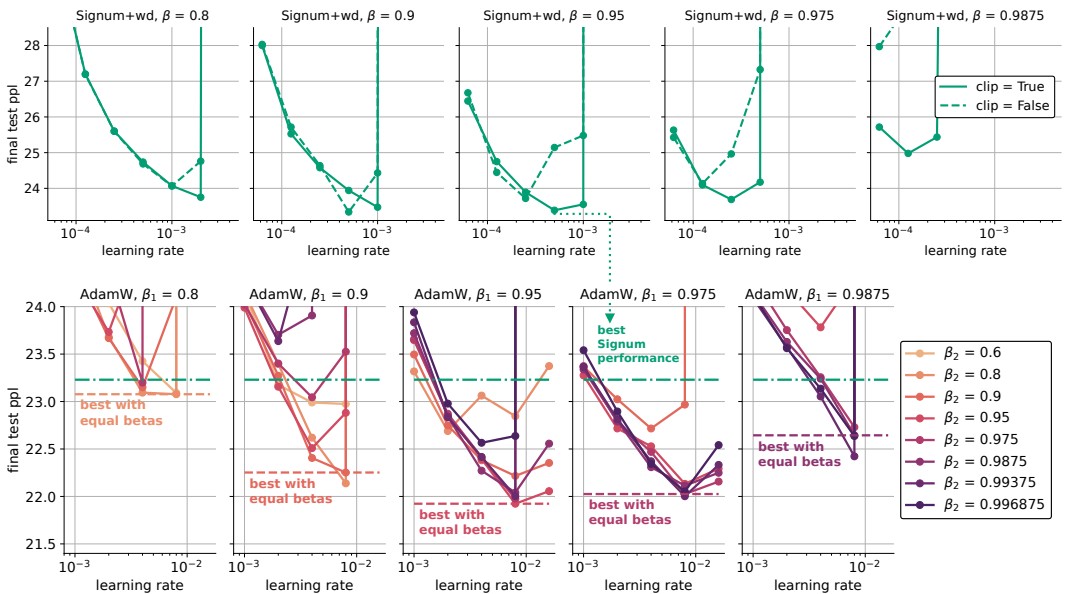

Figure 2: *Training a total of **265 language models** with **160M** parameters with **3.2B** SlimPajama-627B tokens, sequence length of 2048, batch size of 256. Shown is the final test perplexity on 100M held-out tokens. Some underperforming runs are not shown to keep focus on the most interesting range. For a careful description of our tuning grid, see §A. Takeaway 1: Validation perplexity of highly tuned (65 hyperparameter configurations)* `Signum` *with weight decay 0.1 – top row – is around 23.23 (see Table 1 for multiple seeds at optimal tuning). We ablate on the momentum parameter, learning rate, and presence of global clipping before averaging. The best performance of* `Signum` *is reported as a green horizontal line on the second row (200* `Adam` *runs, with weight decay of 0.1). Most* `Adam` *runs perform better than optimally tuned* `Signum`*. Takeaway 2: For each* $\beta_1$*, the optimal corresponding* $\beta_2$ *(after learning rate tuning) is similar. The higher* $\beta_1$*, the higher* $\beta_2$ *for optimal performance (optimal* $\beta$*s are correlated).*

## 2 Preliminaries and Related Works

For a signal $(s_k)_{k \in \mathbb{N}}$ and $\beta \in [0, 1)$, we define the $\beta$-normalized exponential moving average:

$$\mathrm{EMA}_\beta[s_k] = \beta \mathrm{EMA}_\beta[s_{k-1}] + (1 - \beta)s_k, \qquad \mathrm{EMA}_\beta[s_0] := s_0 \text{ (or zero)}. \tag{1}$$

The `Adam` optimizer [Kingma and Ba, 2014] without bias correction [3] takes the following form:

$$w_{k+1} = w_k - \eta_k \left( \sqrt{\mathrm{EMA}_{\beta_2}[g_k^2]} + \epsilon \right)^{-1} \mathrm{EMA}_{\beta_1}[g_k] \tag{Adam}$$

where all division and multiplications are element-wise, $w_k, g_k \in \mathbb{R}^d$ are model parameters and gradients at iteration $k$, $\eta_k$ is the scheduled learning rate, and $\epsilon > 0$ is a small constant. `RMSprop` [Tieleman and Hinton, 2012] is an earlier method that sets $\beta_1 = 0$.

One special case, and simplification, of `Adam` is to consider $\beta_1 = \beta_2 = \epsilon = 0$ which gives `SignSGD`:

$$w_{k+1} = w_k - \eta_k \mathtt{sign}[g_k]. \tag{SignSGD}$$

A practical variant of `SignSGD`, which has shown strong performance in language modeling [Kunstner et al., 2023], first computes an exponential moving average (EMA) – or momentum – of the gradients before applying the `sign` operator [Bernstein et al., 2018]:

$$w_{k+1} = w_k - \eta_k \mathtt{sign}[\mathrm{EMA}_\beta[g_k]]. \tag{Signum}$$

In practice, every language modeling pipeline (see e.g. [Karpathy, 2022]) incorporates some gradient clipping strategy [Pascanu et al., 2013], a component known to stabilize training in the autoregressive

---

[3]We show in Table 3 that the presence of bias correction does not affect our results at the best hyperparameter configuration. However, for all our runs, we use the Pytorch implementation including this factor, for simplicity.

setting and to make gradients more robust to the stochasticity of language [Zhang et al., 2020b]. Global norm clipping (that we abbreviate Gclip), processes gradients fresh out of the backward pass:

$$\texttt{Gclip}[g_k] = \min\left\{1, \frac{1}{\|g_k\|_2}\right\} g_k.$$

In our experiments, we start from vanilla SGD with momentum: $w_{k+1} = w_k - \eta_k \texttt{EMA}_\beta[g_k]$ and ablate on the positive effect of Gclip before applying momentum. Regarding coordinate clipping (Cclip), a softer version of $\texttt{sign}$, we consider applying it to $\texttt{EMA}_\beta[g_k]$ – in connection with $\texttt{Signum}$.

**Research on Adam, a short summary.** Despite initial concerns on generalization [Wilson et al., 2017] and convergence [Reddi et al., 2018], after the introduction of decoupled weight decay (i.e., $\texttt{AdamW}$ [Loshchilov and Hutter, 2019]) $\texttt{Adam}$ rapidly became the de-facto standard optimizer in deep learning, with works highlighting its landscape adaptation properties [Orvieto et al., 2022] and its debated connections to empirical Fisher preconditioning [Kunstner et al., 2019].

With the advent of Transformers [Vaswani et al., 2017], early works noticed an intriguing gap with SGD performance in language modeling [Xiong et al., 2020] (much larger than what can be observed, e.g., in CNNs on image data), that was initially attributed to heavy-tail noise in text data [Simsekli et al., 2019, Zhang et al., 2020a] – suggesting $\texttt{Adam}$ performance to be correlated with its adaptive coordinate clipping mechanism [Zhang et al., 2020a].

As models became larger and more hardware-demanding, interest spiked in the community to reduce the memory footprint of $\texttt{Adam}$ [Li et al., 2023, Zhang et al., 2024b] and to search for more efficient options [Chen et al., 2023, Liu et al., 2023]. Current trends, draw an intriguing connection between $\texttt{Adam}$ and $\texttt{SignSGD}$ [Bernstein and Newhouse, 2024], and in particular with its momentum variant: $\texttt{Signum}$ [Bernstein et al., 2018]. This connection was first suggested in early attempts to understand $\texttt{Adam}$'s empirical performance [Balles and Hennig, 2018], and has recently gained renewed attention in light of transformer architectures and their heterogeneous optimization landscapes [Noci et al., 2022, Zhang et al., 2024a, Tomihari and Sato, 2025, Kunstner et al., 2024, Zhao et al., 2025]. These landscape-based arguments are now more compelling, as recent evidence shows that $\texttt{Adam}$ and signed momentum methods outperform SGD even in deterministic settings [Kunstner et al., 2023].

**Our approach.** Although recent literature highlights many connections between $\texttt{Adam}$ and simpler methods such as $\texttt{Signum}$—which involve fewer hyperparameters, the computational demands of thoroughly studying $\texttt{Adam}$ on small- to medium-scale language models remain prohibitive for most academic optimization researchers. This challenge is amplified by the combinatorial explosion of hyperparameter configurations required for rigorous comparisons. In §3, we aim to provide a comprehensive empirical reference for optimizer performance across a range of language modeling settings. Our key findings are distilled into two main takeaways (Figure 2), which are further supported by theoretical insights in §4.

## 3 Experiments

In our experiments, we systematically explore Transformer-based language models using a nanoGPT [Karpathy, 2022] implementation[4] enhanced by recent advancements such as Rotational Positional Embeddings [Su et al., 2024], RMSNorm normalization [Zhang and Sennrich, 2019], and SwiGLU activation functions [Shazeer, 2020]. We adopt a robust training protocol inspired by successful practices established in large language models like LLaMa [Touvron et al., 2023], GPT-neox [Black et al., 2022], GPT-J [Wang and Komatsuzaki, 2022] and Pythia [Biderman et al., 2023], leveraging techniques including bfloat16 precision, linear warm-up followed by a cosine annealing schedule [Loshchilov and Hutter, 2016], and global gradient norm clipping (unless specified). Our model configurations follow [Biderman et al., 2023] and are presented, alongside a detailed description of all tuning settings and resources, in §A.

### 3.1 Extensive benchmarking at 160M parameters

We conduct 475 compute-optimal pretraining runs on the SlimPajama-627B dataset [Soboleva et al., 2023], using a sequence length of 2048, a batch size of 256, and a decoupled weight decay of 0.1 [Loshchilov and Hutter, 2019] (except for SGD). We always report validation perplexity on a

---

[4]https://github.com/Niccolo-Ajroldi/plainLM/tree/main

held-out subset of 100M tokens. Results from these tuning sweeps are summarized in Table 1, Figure 2, and Appendix B.1. The runs span the following configurations:

- `SGD` (131 runs): Tuned parameters include weight decay (too large causes instability), global norm clipping (Gclip). We also consider clipping coordinates after applying momentum (Cclip). For all these options, momentum and learning rates are independently tuned.

- `RMSprop` (48 runs): Tuned parameters include momentum on the preconditioner and learning rate.

- `Signum` (70 runs): Tuned parameters include global norm clipping, momentum, and learning rate.

- Momentum on `SignSGD` (35 runs): This variant inverts the order of the `sign` and `EMA` operations (and performs worse than `Signum`). Clipping has no effect here due to the sign operation.

- `AdamW` (200 runs): Tuned parameters include both momentum terms and the learning rate.

Two additional seeds are provided for the best performing hyperparameter settings, see Table. 1.

**Choice for betas grid.** While we vary the learning rate by powers of two, our choice of moving average parameters is guided by recent insights into `Adam` scaling behavior [Malladi et al., 2022, Compagnoni et al., 2025]: we choose $\beta = 1 - \kappa(1 - \beta_{\text{base}})$ where $\beta_{\text{base}} = 0.9$ and $\kappa \in \{2^{-5}, 2^{-4}, \ldots, 2^2\}$. This makes it such that the accumulation factor $1/(1 - \beta) = 1/(\kappa(1 - \beta_{\text{base}}))$.

**Takeaway 1**. As shown in Figure 2 and Table 1, optimally tuning `Signum` with weight decay leads to significant improvements over standard `SGD`, in line with recent findings [Kunstner et al., 2023, Zhao et al., 2025]. Nonetheless, `Adam` consistently outperforms the alternatives across most settings, suggesting that it retains a key advantage—a "secret sauce"—that continues to set it apart from better-understood methods in large-scale optimization tasks.

This gap is not limited to this specific setup. In §3.2 we discuss results on another dataset (Fineweb), with disabled weight decay, and shorter sequence lengths. Further, we ablate on other potential confounders (initialization of moving averages, bias corrections, Adam $\epsilon$ value) in §3.3.

**Takeaway 2 (a)**. In Figure 2, we clearly see that $\beta_1 = \beta_2$ yields near-optimal performance in `Adam`, for the five $\beta_1$ values we considered. In § 3.2 we show similar results at different batch sizes, different sequence lengths, and with disabled weight decay on a different dataset. We also extend this observation to 410M parameters models (Figure 5). This empirical finding serves as a basis for our theory in §4.

**Takeaway 2 (b)**. As a corollary to Takeaway 2, Figure 3 shows that the best performance is not only achieved when $\beta_1 = \beta_2$, but also improves as the two values become closer. Among 500 runs on 160M-parameter models, we observe a clear correlation: lower loss is associated with smaller differences between $\beta_1$ and $\beta_2$. This suggests that gradient smoothing ($\beta_1$) and preconditioner smoothing ($\beta_2$) should not be treated as independent operations—optimal performance often arises when they act in concert.

To put to the test our second takeaway in **different training settings**, we consider shorter sequence lengths (512, Figure 14), higher/lower batch sizes (Figure 16 & Figure 17), different data (Fineweb) and absence of weight decay (Fig, 18). See discussion in §3.2.

**Standard choice for betas.** While in standard deep learning (also Pytorch default) $\beta_2 > \beta_1$ (0.999, 0.9), in language modeling the choice $\beta_1 = 0.9$, $\beta_2 = 0.95$ is much more common. A lower value for $\beta_2$ was shown to help mitigate loss spikes [Cattaneo and Shigida, 2025, Compagnoni et al., 2025], and several recent studies have started to adopt $\beta_1 = \beta_2 = 0.95$ as a default [Zhao et al., 2025, Shah et al., 2025, Zhang et al., 2025]. All our findings confirm this choice for tuning (see e.g. Figure 2), of which we evaluate validity extensively for several values of $\beta_1$.

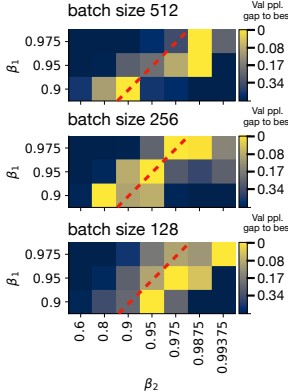

Figure 3: *Summary of the results in §B.4. At different batch sizes, for each $\beta_1 \in [0.9, 0.95, 0.975]$, we show the best-performing $\beta_2$ (highest score, yellow) and the gap between its performance and that of other options in the grid. We notice high correlation between beta values (e.g., $\beta_2 = 0.9875$ is a terrible option at $\beta_1 = 0.9$, but a good one at $\beta_1 = 0.975$). While results are noisy, notice that $\beta_1 = \beta_2$ never degrades performance more than 0.3 points. In contrast (Table 1, the gap with `Signum` can be as high as 1.37 points.*

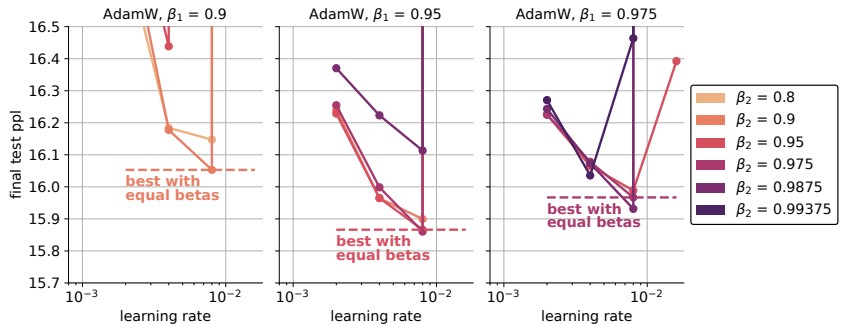

Figure 5: *The final validation performance (100M held-out tokens) for 44 trained LMs with 410M parameters trained on 8.2 B SlimPajama tokens (Chinchilla-optimal).* **Equal betas yields near-optimal performance**. *We use gradient clipping and a batch size of 512 (scaled by 2 compared to Figure 2, as suggested by Zhang et al. [2025]). Sequence length is 2048, weight decay is 0.1. Note that the standard setting (0.9, 0.95) is quite suboptimal here.*

**Theoretical relations between betas.** We note that a correlation between $\beta$ parameters was also noted first by Reddi et al. [2018], Alacaoglu et al. [2020] for AMSgrad, and later by Zhang et al. [2022] for Adam, where it is shown that if $\beta_2$ is large enough and $\beta_1 < \sqrt{\beta_2}$, it converges to the neighborhood of critical points. Further, Xie and Li [2024] showed that weight decay in AdamW leads to convergence to a constrained minimizer only if $\beta_2 > \beta_1$.

## 3.2 Ablations

**More Tokens.** We find our Takeaway 2 to also hold at a higher token budget. In §B.2, we show a trend very similar to Fig. 2 for models trained for $2\times$ the Chinchilla-optimal budget.

**Different batch size.** We find our Takeaway 2 to be robust to batch size. In the same setting as Figure 2 yet at a slightly lower compute budget due to hardware limitations (2.5B parameters), we find that, even at batch size 128 and 512 the choice $\beta_1 = \beta_2$ yields near-optimal performance. This step involves training 500 models, see §B.4 for visualizations similar to Figure 2 and a discussion.

**Different sequence length.** In §B.3, we find our Takeaway 2 to also hold at shorter sequence length of 512 (Figure 14). We note that here performance of Signum is closer to that of Adam compared to Figure 2 – yet, Adam is still superior by a substantial margin ( 0.7 validation perplexity), Takeaway 1. This pattern agrees well with the results in [Zhao et al., 2025], who found other methods to be competitive with Adam at short context lengths. Our experiments in Figure 14 and Figure 2 suggest that Adam performance particularly shines at higher sequence lengths.

**Different data and weight decay.** In Figure 18 we test both Takeaway 1 and Takeaway 2 on Fineweb [Penedo et al., 2024]. We take this opportunity to also deactivate weight decay ($\lambda = 0$), as the optimal Signum learning rates in Figure 2 suggest decoupled weight decay $w = w - \lambda\eta w$ acts differently for the two methods, likely needing different tuning. When deactivated, we still see a substantial gap between Signum and Adam, as well as strong performance with equal betas.

**Larger Models.** We restrict our attention to the SlimPajama dataset and to validation of Takeaway 2. Results are presented in Figure 5, comprising 44 full compute-optimal training runs of 410M parameter models, which confirm yet again strong and near-optimal performance at $\beta_1 = \beta_2$.

## 3.3 Checking for confounders

When comparing Signum with Adam, here are a few confounders that might affect results:

**The value of $\epsilon$ in Adam** was shown to be important for numerical stability, and might affect performance [Yuan and Gao, 2020]. We show in Table 2 that one can choose an extremely small $\epsilon$

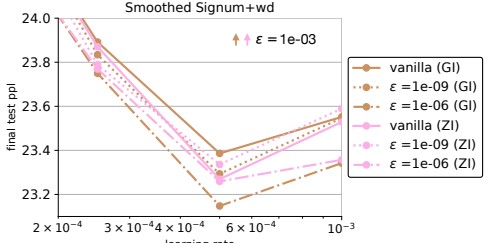

Figure 4: *Adding an $\epsilon$ mollifier to Signum, i.e., using $m_k/(\sqrt{m_k^2} + \epsilon)$ offered little to no improvement. We also test both zero initialization (ZI) and gradient initialization (GI) for $m$, and find similar results with no significant improvement. $\epsilon = 1e - 3$ is significantly worse, hence is not shown. Similar finding: Figure 7.*

value in our setting. We cross-check the impact of including an $\epsilon$ factor in Signum: we found that

little can be gained from this strategy (Figure 4). In short, we found that $\epsilon$ is not a crucial parameter in our setup. This is also liked to our findings on adaptive mollifiers, cf. §4.

**Initialization of moving average parameters.** In Figure 4 we also ablate on initialization of the moving average in `Signum` and found no substantial differences. We perform this same ablation for `Adam` and report comprehensive results with seeds in §B.6.

**Bias correction.** While bias correction in `Adam` is helpful in early training, final validation performance is almost unchanged, see the full training curve and results with seeds in §B.6.

Table 2: *Effect of $\epsilon$ in* `AdamW`*– other parameters optimally tuned for $\epsilon = 10^{-8}$ (setting: Figure 2). All values between $\epsilon \in [10^{-6}, 10^{-15}]$ result in a similar validation perplexity.*

|  | $\epsilon = 1e-3$ | $\epsilon = 1e-6$ | $\epsilon = 1e-8$ | $\epsilon = 1e-10$ | $\epsilon = 1e-12$ | $\epsilon = 1e-15$ |
|---|---|---|---|---|---|---|
| Val ppl | 23.34± 0.31 | 21.56± 0.19 | 21.86± 0.21 | 21.87± 0.04 | 21.89± 0.2 | 21.91± 0.18 |

## 4 New Viewpoints of `Adam`

We now show that restricting to the case $\beta_1 = \beta_2 = \beta$ yields a useful interpretation of `Adam`. Since the `Adam` update is coordinate-wise, it suffices to analyze a single scalar gradient $g_k \in \mathbb{R}$. Moreover, ablations (Table 2, Table 3) indicate that neither the $\epsilon$-term nor the bias correction significantly affect performance. Thus, for clarity, we set $\epsilon = 0$ and study the simplified `Adam` update:

$$d_k = \frac{\text{EMA}_\beta[g_k]}{\sqrt{\text{EMA}_\beta[g_k^2]}}. \tag{2}$$

We next rewrite (proof in the Appendix) the update to explicitly highlight the role of variance.

**Proposition 1.** *Let $m_k = \text{EMA}_\beta[g_k]$. Then the update (2) admits the equivalent representation:*

$$d_k = \frac{m_k}{\sqrt{m_k^2 + \beta \, \text{EMA}_\beta[(m_{k-1} - g_k)^2]}}. \tag{3}$$

This shows that the denominator depends on the exponential moving average of the squared deviation between the momentum $m_{k-1}$ and the incoming gradients $g_k$, with an **interesting multiplier** $\beta$. As we demonstrate in the next section, this quantity is in fact an online estimator of the gradient variance.

### 4.1 `Adam` **Estimates Mean and Variance using Variational Inference**

We show that `Adam` admits a natural interpretation as an online variational inference method, where

$$m_k := \text{EMA}_\beta[g_k] \quad \text{and} \quad \sigma_k^2 := \beta \, \text{EMA}_\beta[(m_{k-1} - g_k)^2]$$

correspond to online estimates of the mean and variance of the stochastic gradients. We reintroduce `Adam` through this lens.

Suppose we are given a sequence of stochastic gradients $\{g_1, \ldots, g_k\}$, where each $g_k$ is sampled from an unknown Gaussian distribution whose mean and variance may vary with $k$. Rather than taking steps directly along these noisy gradients, we aim to estimate their mean and variance online and use these estimates to define a more informed search direction.

At iteration $k$, let $(m_k, \sigma_k^2)$ denote our current estimates of the gradient mean and variance, respectively. Upon receiving a new gradient sample $g_{k+1} \sim \mathcal{N}(m, \sigma^2)$ with unknown $(m, \sigma^2)$, we wish to update our estimates to $(m_{k+1}, \sigma_{k+1}^2)$ so that it becomes more *likely* that $g_{k+1}$ was drawn from $\mathcal{N}(m_{k+1}, \sigma_{k+1}^2)$. Since we also expect the underlying distribution to vary slowly over time, we prefer that $\mathcal{N}(m_{k+1}, \sigma_{k+1}^2)$ remain close to the previous estimate $\mathcal{N}(m_k, \sigma_k^2)$. These two goals—fitting the new observation and ensuring smooth updates—can be traded off via a regularized maximum likelihood problem:

$$\min_{m, \sigma^2 \geq 0} -\log p(g_{k+1} \mid m, \sigma^2) + \tfrac{1}{\lambda} \text{KL} \left( \mathcal{N}(m_k, \sigma_k^2) \,\|\, \mathcal{N}(m, \sigma^2) \right), \tag{4}$$

where $p(g_{k+1} \mid m, \sigma^2)$ is the Gaussian likelihood, $\lambda \geq 0$ is a regularization parameter, and KL denotes the Kullback–Leibler divergence:

$$-\log p(g_{k+1} \mid m, \sigma^2) = \frac{1}{2}\log \sigma^2 + \frac{1}{2\sigma^2}(g_{k+1} - m)^2, \tag{5}$$

$$\mathrm{KL}\left(\mathcal{N}(m_k, \sigma_k^2) \,\|\, \mathcal{N}(m, \sigma^2)\right) = \frac{1}{2}\left[\frac{\sigma_k^2}{\sigma^2} + \frac{(m_k - m)^2}{\sigma^2} - 1 - \log\left(\frac{\sigma_k^2}{\sigma^2}\right)\right]. \tag{6}$$

The following result, proved in the appendix, characterizes the solution of (4), showing that the moving averages used in `Adam` correspond exactly to an instance of online variational inference:

**Theorem 4.1.** Let $\beta = \frac{1}{1+\lambda}$. Then the solution to the optimization problem (4) is given by

$$m_{k+1} = \beta m_k + (1 - \beta)g_{k+1} = \mathtt{EMA}_\beta[g_{k+1}], \tag{7}$$

$$\sigma_{k+1}^2 = \beta\sigma_k^2 + \beta(1-\beta)(m_k - g_{k+1})^2 = \beta\,\mathtt{EMA}_\beta\left[(m_k - g_{k+1})^2\right]. \tag{8}$$

As a consequence, the `Adam` update direction in (3) can be rewritten as

$$d_k = \frac{m_k}{\sqrt{m_k^2 + \beta\mathtt{EMA}_\beta[(m_{k-1} - g_k)^2]}} = \frac{m_k}{\sqrt{m_k^2 + \sigma_k^2}} = \frac{\mathrm{sign}(m_k)}{\sqrt{1 + \sigma_k^2/m_k^2}}. \tag{9}$$

This shows that `Adam` may be interpreted as an *adaptive mollified* variant of `Signum`, where the mollification depends on the local noise-to-signal ratio. This mollified viewpoint aligns well with one of the first papers on understanding `Adam` [Balles and Hennig, 2018], as discussed after Proposition 1.

Using these insights, we can better formalize the *noise-to-signal* interpretation of `Adam` [Balles and Hennig, 2018] (see also §4.2). Let $m_k/\sigma_k$ denote the signal-to-noise ratio (SNR). We show that `Adam` can be viewed as a steepest descent method whose trust region is modulated by the SNR.

To build this connection, consider first the `Signum` update. It corresponds to the steepest descent direction under an $\ell_\infty$-norm constraint [Balles and Hennig, 2018], solving

$$-\mathrm{sign}(m_k) = \underset{\theta \in \mathbb{R}}{\arg\min} \; -m_k \cdot \theta \quad \text{subject to } |\theta| \leq 1. \tag{10}$$

That is, `Signum` selects the direction most aligned with $-m_k$ within a unit trust region.

In contrast, `Adam` can be interpreted as a steepest descent method with a variable trust region, defined by the (inverse) signal-to-noise ratio:

$$-\frac{\mathrm{sign}(m_k)}{\sqrt{1 + \sigma_k^2/m_k^2}} = \underset{\theta \in \mathbb{R}}{\arg\min} \; -m_k \cdot \theta \quad \text{subject to } |\theta| \leq \frac{1}{\sqrt{1 + \sigma_k^2/m_k^2}}. \tag{11}$$

Here, the effective step size shrinks when the noise dominates the signal, and expands toward 1 as uncertainty decreases. In this sense, `Adam` adapts its update magnitude according to a confidence-weighted trust region.

## 4.2 Comparison with Balles and Hennig [2018]

Balles and Hennig [2018] first drew a connection between `Adam`, `Signum` and Signal-to-noise Ratio regularization. Their observation was as follows. Let $m_k = \mathtt{EMA}_{\beta_1}[g_k]$, and $v_k = \mathtt{EMA}_{\beta_2}[g_k^2]$. We can trivially re-write the `Adam` direction as

$$d_k = \frac{m_k}{\sqrt{v_k}} = \frac{m_k}{\sqrt{m_k^2 + v_k - m_k^2}}.$$

If we now *assume* that $\sigma_k^2 := v_k - m_k^2$ is a measure of variance, then dividing the `Adam` direction through by $|m_k|$, as done in (9), we arrive at a Signal-to-noise Ratio regularized variant of the `Signum` method. In particular, as the noise goes to zero ($\sigma_k^2 \to 0$), we arrive at the `Signum` method.

The missing piece in their insight was to show when and if the term $v_k - m_k^2$ is a measure of variance.

We show that $\beta_1 = \beta_2$, a choice that was not common[5] at the time of Balles and Hennig [2018], allows for more precise claims: Proposition 1 shows that when $\beta_1 = \beta_2 = \beta$ the term $v_k - m_k^2$ is

---

[5]Default parameters have for long been $\beta_1 = 0.9$, $\beta_2 = 0.999$, see https://docs.pytorch.org/docs/stable/generated/torch.optim.Adam.html.

precisely equal to $\beta\text{EMA}_\beta[(m_{k-1} - g_k)^2]$, which in turn is a online estimate of variance (Theorem 4.1). We further show that $v_k - m_k^2$ only has a precise variance interpretation for the case $\beta_1 = \beta_2$: indeed, we prove in §C.2 that Adam can be represented as

$$d_k = \frac{m_k}{\sqrt{m_k^2 + \gamma \, \text{EMA}_\tau[(am_{k-1} - bg_k)^2]}} \tag{12}$$

for some $a, b, \gamma \in \mathbb{R}$ and $\tau \in (0, 1)$ *if and only if* $\beta_1 = \beta_2$. In other words, connecting $v_k - m_k^2$ to variance estimation, and in turn Adam to an SNR-controlled trust region method (11), can only be done precisely for the case of equal betas.

**Ablating hyperparameters in our reformulation.** While (12) reduces to Adam with equal betas if and only if $a, b = 1$ and $\beta = \gamma = \tau$, we found it interesting to consider (12), with $a = b = 1$, as a new method with no precise connection to simultaneous variance and mean estimation, with hyperparameters $\beta, \gamma, \tau$. In §C.4, we train 150 additional language models ablating on such parameters, and found no advantage in setting $\beta \neq \tau$ or $\tau \neq \gamma$. We believe such evidence further strengthens our claims: best performance is aligned to the theoretical choice $\tau = \gamma = \beta$.

## 5 Why an adaptive trust region? Insights from heterogeneous quadratics

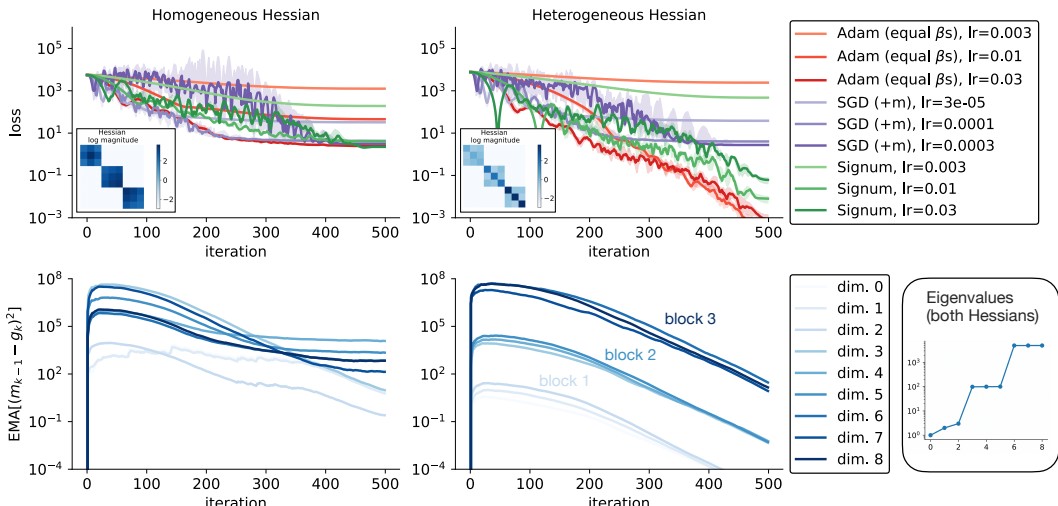

Figure 6: ***Top row:*** *Training performance (median and 25%/75% quantiles over 10 seeds) of SGD, Signum, and Adam on two 9-dimensional convex quadratic problems (§D) inspired by Zhang et al. [2024a]. All optimizers use moving average parameters set to 0.95, with a 10% warmup followed by cosine decay to zero. Both problems share the same Hessian eigenspectrum and have a $3 \times 3$ block structure. The landscape on the left is homogeneous, with each block containing both large and small eigenvalues. The landscape on the right is heterogeneous, with each block having eigenvalues of different magnitudes. In this setting, Adam clearly outperforms SGD, with Signum closing part of the gap.* ***Bottom row:*** *Dynamics of the variance term in Proposition 1. The value of this term varies both across iterations and across blocks, adapting to the local curvature structure. This adaptive behavior improves performance over Signum in the heterogeneous setting.*

While our theoretical analysis in §4 offers a new perspective on Adam, it is not tied to any specific architecture. To enhance intuition and provide a controlled setting for future work, we validate our findings on a simplified model of transformer loss landscapes introduced by Zhang et al. [2024a], building on signal propagation theory [Noci et al., 2022].

As noted in Zhang et al. [2024a], Kunstner et al. [2024], Zhao et al. [2025], the landscape of autoregressive language models is highly heterogeneous: Hessian blocks associated with semantically distinct parameter groups (e.g., normalization layers, embeddings, or softmax-related parameters) exhibit markedly different eigenspectra and thus demand different learning rates. In contrast to homogeneous models (e.g., CNNs), this heterogeneity is where Adam significantly outperforms SGD [cf. Zucchet and Orvieto, 2024].

Figure 6 illustrates this point. On a toy heterogeneous quadratic landscape, tuned Adam with equal $\beta$ values substantially outperforms tuned SGD with momentum, echoing results from Zhang et al.

[2024a]. We also observe that `Signum` closes much of the gap but still falls short of `Adam`. This is consistent with our findings in Table 1 for language models.

In Proposition 1, we showed that the key difference between `Signum` and `Adam` lies in the variance correction term $\beta\mathrm{EMA}_\beta[(m_{k-1} - g_k)^2]$ in the denominator. Understanding how this term evolves is essential: it cannot be approximated by a constant. In the second row of Figure 6, we observe that the variance estimate not only varies over time, but also differs in scale across the three blocks—mimicking the parameter groupings in transformer models. This block-wise variation reinforces the idea that the variance term dynamically adapts to the local curvature and cannot be substituted by a fixed value. In Figure 7 and 4, we show a similar effect in heterogeneous quadratic and language models, respectively: replacing $\beta\mathrm{EMA}_\beta[(m_{k-1} - g_k)^2]$ with a fixed constant $\epsilon$ cannot provide the same adaptive effect.

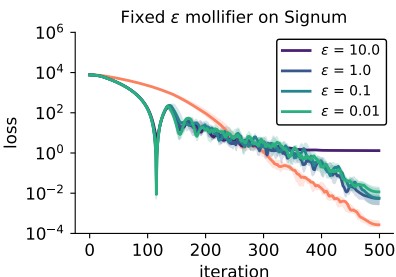

Figure 7: *Counterpart of Figure 4 for the heterogeneous quadratic example. We do not observe gains with a fixed mollifier $m_k/\sqrt{m_k^2 + \epsilon}$. Placing inside or outside $\sqrt{\cdot}$ has no qualitative effect after tuning.*

## 6 Conclusion

We have presented an extensive numerical study of `Adam`, comparing it against several proposed simplifications. Our main finding is that, on generative language modeling tasks, `Adam` significantly outperforms these simplified variants. Notably, we observe that setting $\beta_1 = \beta_2$ is often optimal or near-optimal. Based on this observation, we recommend `Adam` with $\beta_1 = \beta_2$ as a simplified model, and we provide a new variational inference interpretation for this setting.

Our findings come with some limitations. First, our numerical experiments fix a grid over the hyperparameters; the results are therefore sensitive to the choice of grid, and different grids may lead to different conclusions. However, for all our hyperparameters, we show explicitly all tuning curves demonstrating that we are always at optimality inside the grid (and not at the edge). Second, while $\beta_1 = \beta_2$ often performs well, we note that at small batch sizes, Figure 3 suggests a slight shift. Finally, although Theorem 4.1 shows that `Adam`'s two momentum buffers can be interpreted as online estimates of the gradient's mean and variance, it does not explain why these estimates should be arranged into the quotient used in `Adam` (9). Lemma 1 in [Balles and Hennig, 2018] can provide a starting point to further dissect this interesting choice and explore alternatives.

## Acknowledgements

We would like to thank Niccolo Ajroldi, Sam Liang, Weronika Ormaniec, and Enea Monzio Compagnoni for their comments. We additionally thank the NeurIPS 2025 and ICML 2025 HiLD workshop reviewers for their valuable feedback and references. Antonio Orvieto acknowledges the financial support of the Hector Foundation, and is thankful for the compute made available by MPI-IS and the Tübingen AI ecosystem.

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

# Contents

# A   Experimental details

For pre-training Transformers on Causal Language Modeling, we build upon the nanoGPT [Karpathy, 2022] implementation, augmenting it with Rotational Positional Embedding [Su et al., 2024], RM-SNorm [Zhang and Sennrich, 2019], and SwiGLU [Shazeer, 2020]. All our models have a vocabulary size of 50280 and make use of GPT-Neox tokenizer [Black et al., 2022]. We adopt an enhanced training recipe, made popular by large language models such as LLaMa [Touvron et al., 2023]. These modifications include: training in bfloat16; employing a linear learning rate warm-up for 10% of the training steps, followed by cosine annealing to $1e - 5$. Global norm clipping is used (unless specified or ablated upon) for gradients with norms above 1 (on the raw gradient, as a first step). We have no weight tying between the embedding and the last linear layer. We always report validation perplexity on a separate subset consisting of 100M tokens. Seeds, when provided, are relative to distinct network initialization.

**Computational Resources.**   All our experiments at a 160M parameter scale are performed on a single NVIDIA A100-SXM4-80GB. At compute optimality (most of our experiments) each run takes approximately 5.83 hours. Our runs at a 410M parameter scale are performed on 8 NVIDIA A100-SXM4-80GB GPUs, and each run here takes approximately 4.83 hours. For all our runs, we fill up memory and optimize to minimize the gradient accumulation steps (usually, around 8).

**Code.**   All our runs use the repository

<p align="center">https://github.com/Niccolo-Ajroldi/plainLM</p>

**Model settings (160M).**   We use the same configuration as [Biderman et al., 2023]: `https://github.com/EleutherAI/pythia/blob/main/models/160M/pythia-160m.yml`

- *Layers:* 12 Transformer [Vaswani et al., 2017] layers
- *Attention heads:* 12
- *Hidden size:* 768
- *Attention implementation:* Flashattention [Dao et al., 2022].
- *MLP type:* SwiGLU [Shazeer, 2020] with expansion factor 8/3.
- *Backbone:* PreLN transformer [Xiong et al., 2020] with skip connections.
- *Normalization:* RMSnorm [Zhang and Sennrich, 2019] for both Attention and MLP.
- *Position embeddings:* Rotary embeddings (RoPE) to 25% of dimensions ([Su et al., 2024])
- *Initialization:* the MLP and Attention output weights are initialized with variance $0.02/\sqrt{2\#\text{layers}}$ (scaling also similar to [Radford et al., 2019]). All other weights (comprising embeddings) are initialized with a standard deviation of 0.02 (Nguyen and Salazar [2019], Wang and Komatsuzaki [2022], Sec. 2.2). Biases are always initialized at zero.
- *Precision:* Mixed precision FP16 enabled.
- *Dropout:* Disabled for both hidden and attention layers (see also Chowdhery et al. [2023]).

**Model settings (410 M).**   We use the same setting as [Biderman et al., 2023], configuration can be found here: `https://github.com/EleutherAI/pythia/blob/main/models/410M/pythia-410m-deduped.yml`

- *Layers:* 24 Transformer layers
- *Attention heads:* 16
- *Hidden size:* 1024
- *Other settings as 160M parameters.

## A.1 Experiments on SlimPajama – 160M parameters model

On the Cerebras SlimPajama-627B [Soboleva et al., 2023] dataset: `https://huggingface.co/datasets/cerebras/SlimPajama-627B` at a 160M scale we present three experimental sections:

- Section A.1.1 – core setting, ablating on **all optimizers**.
- Section A.1.3 – ablating on a **smaller sequence length**.
- Section A.1.4 – ablating at **different batch sizes**.

### A.1.1 Sequence Length 2048, Batch size 256, 3.2 B Tokens (6200 gradient steps)

This setup comprises a total of 747 full training runs. We always use warm-up (10%) and cosine anneal until a learning rate of $1e-5$. This setting is Chinchilla-optimal (20 tokens/parameter).

$\lambda$ here denotes the weight decay, always decoupled [Loshchilov and Hutter, 2019].

**Core experiments:** These are the core experimental results for this setting.

- **SGD** with momentum $\beta$ and **global norm clipping** to 1 (Gclip, dampening to $1-\beta$) — *84 full runs* (Figure 8, top).

$$(\eta, \beta, \lambda) \in [2.0, 1.0, 0.5, 0.25, 0.125, 0.0625, 0.03125]$$
$$\times [0.9875, 0.975, 0.95, 0.9]$$
$$\times [0, 1e-3, 1e-4].$$

- **SGD** with momentum $\beta$ with (1) **global norm clipping** of raw gradient to 1 (Gclip) and (2) **coordinate clipping** (Cclip) to 1 after momentum is applied. Dampening is set to $1-\beta$, $\lambda$ (weight decay) is set to 0, as the previous point revealed decreasing performance on SGD — *24 full runs* (Figure 8, bottom).

$$(\eta, \beta, \lambda) \in [2.0, 1.0, 0.5, 0.25, 0.125, 0.0625]$$
$$\times [0.9875, 0.975, 0.95, 0.9]$$

- **SGD** with momentum $\beta$ (vanilla, dampening to $1-\beta$, **no clipping**). $\lambda = 0$ (weight decay). — *28 full runs* (Figure 9)

$$(\eta, \beta) \in [0.25, 0.125, 0.0625, 0.03125, 0.015625, 0.0078125, 0.00390625]$$
$$\times [0.9875, 0.975, 0.95, 0.9].$$

- **Adam** with global norm clipping to 1 and with $\lambda = 0.1$ (weight decay) and $\epsilon = 1e-8$ (usual Pytorch setup, see also Biderman et al. [2023]).
  – *200 full runs* (Figure 2)

$$(\eta, \beta_1, \beta_2) \in [0.016, 0.008, 0.004, 0.002, 0.001]$$
$$\times [0.9875, 0.975, 0.95, 0.9, 0.8]$$
$$\times [0.996875, 0.99375, 0.9875, 0.975, 0.95, 0.9, 0.8, 0.6]$$

- **Adam without global norm clipping** and with $\lambda = 0.1$ (weight decay) and $\epsilon = 1e-8$ (usual Pytorch setup, see also Biderman et al. [2023]).
  – *165 full runs* (Figure 12)

$$(\eta, \beta_1, \beta_2) \in [0.032, 0.016, 0.008, 0.004, 0.002, 0.001]$$
$$\times [0.975, 0.95, 0.9, 0.8]$$
$$\times [0.9875, 0.975, 0.95, 0.9, 0.8, 0.6]$$

- **RMSprop** implemented using the AdamW Pytorch class using $\beta_1 = 0$. We again use $\lambda = 0.1$ (weight decay) and $\epsilon = 1e-8$.
  – *48 full runs* (Figure 10).

$$(\eta, \beta_2) \in [0.004, 0.002, 0.001, 0.0005, 0.00025, 0.000125]$$
$$\times [0.9875, 0.975, 0.95, 0.9, 0.8, 0.6, 0.4, 0.0]$$

- **Signum** with weight decay $\lambda = 0.1$ as also suggested by [Zhao et al., 2025] (their Figure 5, top left panel). We **ablate on presence of global norm gradient clipping** (to norm 1).
  – *70 full runs* (Figure 2).

$$(\eta, \beta, \text{clip}) \in [0.004, 0.002, 0.001, 0.0005, 0.00025, 0.000125, 0.0000625]$$
$$\times [0.9875, 0.975, 0.95, 0.9, 0.8]$$
$$\times [\text{True, False}]$$

Note that Signum with and without gradient clipping are two different methods: here, clipped gradients are first averaged and only then the sign is taken. Instead, clipping on the EMA of signed gradients (next method) should have no effect (apart from non-determinism).

- **EMASign** with weight decay $\lambda = 0.1$. We ablate on the presence of global norm gradient clipping (to norm 1) *out of mistake*: the two methods are equal!
  – *70 runs (35 duplicate runs)* (Figure 11)

$$(\eta, \beta, \text{clip}) \in [0.001, 0.0005, 0.00025, 0.000125, 0.0000625, 0.00003125, 0.000015625]$$
$$\times [0.9875, 0.975, 0.95, 0.9, 0.8]$$
$$\times [\text{True, False}]$$

**Ablations:**  These ablations were performed to test side-claims in the paper.

- **Adam** with global norm clipping to 1 and $\lambda = 0.1$, $\beta_1 = \beta_2 = 0.95$, $\eta = 0.008$ (best setup from Figure 2). We report performance for 3 seeds using different $\epsilon$ values.
  – *18 full runs* (Table 2).

$$\epsilon \in [1e-3, 1e-6, 1e-8, 1e-10, 1e-12, 1e-15],$$

and influence of initializing exponential moving averages to zero (default, ZI) or to the stochastic quantity of interest (gradient initialization, GI). At the same time, we try to remove bias correction. These experiments are presented with 3 random initialization seeds:
  – *9 full runs* (Table 3).

- **Signum** with global norm clipping to 1 and $\lambda = 0.1$, $\beta = 0.95$ (best setting from Figure 2): we ablate on fixed mollifiers for zero-initialized (ZI) or gradient-initialized (GI) momentum. The mollified we study is $m_k/(\sqrt{m_k} + \epsilon)$:

$$(\eta, \epsilon) \in [0.001, 0.0005, 0.00025, 0.000125] \times [1e-3, 1e-6, 1e-9]$$

  – *12 full runs* (Table 2).
  We additionally test the influence of ZI vs. GI with three random seeds at $\epsilon = 0$.
  – *5 full runs* (Table 3).

**Other:**  for the best-performing variants of core experiments, we initialize the model with two other random seeds. This accounts for
– *14 additional full runs* (Table 1).

### A.1.2  Sequence Length 2048, Batch size 256, 6.4 B Tokens  (12400 gradient steps)

The setup here is exactly as in §A.1.1, but we train for $2\times$ the token budget. We test our core claim ($\beta_1 = \beta_2$ works well), and hence we run:

- **Adam** with global norm clipping to 1 and with $\lambda = 0.1$ (weight decay) and $\epsilon = 1e-8$.
  – *168 full runs* (Figure 13)

$$(\eta, \beta_1, \beta_2) \in [0.032, 0.016, 0.008, 0.004, 0.002, 0.001, 0.0005]$$
$$\times [0.9875, 0.975, 0.95, 0.9]$$
$$\times [0.99375, 0.9875, 0.975, 0.95, 0.9, 0.8]$$

### A.1.3  Sequence Length 512, Batch size 256, 3.2 B Tokens  (24800 gradient steps)

This setup comprises a total of 55 full training runs. We test our core claims (Signum underperforms Adam, $\beta_1 = \beta_2$ works well) at a smaller sequence length. Setting is exactly the same as §A.1.1 for all methods, unless stated otherwise.

- **Adam**, we limit this ablation to $\beta_1 = 0.95$,

  $(\eta, \beta_2) \in [0.001, 0.002, 0.004, 0.008, 0.016] \times [0.99375, 0.9875, 0.975, 0.95, 0.9, 0.8]$

  – *25 full runs* (Figure 14).
- **Signum**, we do a full ablation using global norm gradient clipping to 1.

  $(\eta, \beta) \in [0.0000625, 0.000125, 0.00025, 0.0005, 0.001, 0.002] \times [0.9875, 0.975, 0.95, 0.9, 0.8]$

  – *30 full runs* (Figure 14).

### A.1.4 Sequence Length 2048, Variable batch size, 2.5 B Tokens

We use here a slightly reduced token budget (2.5B, 20 tokens for every non-embedding parameter) and run the same Adam tuning experiment presented in Figure 2 for batch size 256. We actually run this experiment again at a batch size of 256, and test batch sizes of 128 and 512 reducing or doubling the number of steps accordingly (same token budget). The sequence length is still 2048, and the dataset SlimPajama. Due to the reduced number of tokens, each run takes approximately 4.7 hours on our hardware. We implement variation of batch size using gradient accumulation $(4, 8, 16)$ at a micro-batch size of 32 sequences. This setup comprises a total of 500 full training runs.

**Adam** with $\lambda = 0.1$ (weight decay) and $\epsilon = 1e - 8$ (usual setup, see Biderman et al. [2023]), we clip gradients to global norm 1.

- For batch size 256:

$$(\eta, \beta_1, \beta_2) \in [0.016, 0.008, 0.004, 0.002, 0.001]$$
$$\times [0.9875, 0.975, 0.95, 0.9, 0.8]$$
$$\times [0.996875, 0.99375, 0.9875, 0.975, 0.95, 0.9, 0.8, 0.6]$$

- For batch size 128 and 512:

$$(\eta, \beta_1, \beta_2) \in [0.0005, 0.001, 0.0014, 0.002, 0.0028, 0.004, 0.0056, 0.008, 0.0112, 0.016]$$
$$\times [0.975, 0.95, 0.9]$$
$$\times 1 - [4, 2, 1, 0.5, 0.25] \cdot (1 - \beta_1) \qquad \text{(i.e. 3 higher and 2 lower values in grid)}$$

Note that here we overturned the learning rate, the reason for this is the square root scaling law in Malladi et al. [2022], Compagnoni et al. [2025]: if batch size scales by 2, learning rate should scale as $\sqrt{2}$. We see in §B.4 that this indeed seems to hold true, yet noise prevents us from making precise verification claims.

– *500 full runs* (§B.4).

### A.2 Experiments on SlimPajama – 410M parameters model, 8.2 B tokens

All our experiments here use the Cerebras SlimPajama-627B [Soboleva et al., 2023] dataset: `https://huggingface.co/datasets/cerebras/SlimPajama-627B`. We focus on evaluating whether $\beta_1 = \beta_2$ yields good performance in this settings. We scale up the batch size by a factor 2 compared to Section A.1, as suggested by [Zhang et al., 2025]. We perform our experiments at compute optimality (8.2B tokens, 20 tokens per parameter).

**Adam** with $\lambda = 0.1$ (weight decay) and $\epsilon = 1e - 8$ (usual setup, see Biderman et al. [2023]), we clip gradients to global norm 1:

- $\beta_1 = 0.9$

$$(\eta, \beta_2) \in [0.016, 0.008, 0.004, 0.002] \times [0.95, 0.9, 0.8]$$

- $\beta_1 = 0.95$

$$(\eta, \beta_2) \in [0.016, 0.008, 0.004, 0.002] \times [0.9875, 0.975, 0.95, 0.9]$$

- $\beta_1 = 0.975$

$$(\eta, \beta_2) \in [0.016, 0.008, 0.004, 0.002] \times [0.99375, 0.9875, 0.975, 0.95]$$

– *44 full runs* (Figure 5).

## A.3 Experiments on Fineweb – 160M parameters model, 3.2B tokens – no weight decay

While testing our claims on a different dataset, we also crucially *remove weight decay* here. Our setting is otherwise identical to that of §A.1.1: Sequence length is 2048, batch size is 256, model has 160 parameters and we train on 3.2B tokens from Fineweb [Penedo et al., 2024] https://huggingface.co/datasets/HuggingFaceFW/fineweb.

- **Adam** with $\lambda = 0$ (no weight decay!) and $\epsilon = 1e - 8$ (usual setup, see Biderman et al. [2023]). We clip gradients to global norm 1.

$$(\eta, \beta_1, \beta_2) \in [0.032, 0.016, 0.008, 0.004, 0.002, 0.001]$$
$$\times [0.975, 0.95, 0.9]$$
$$\times [0.9875, 0.975, 0.95, 0.9, 0.8]$$

    – *90 full runs* (Figure 18)
- **Signum** with $\lambda = 0$ (no weight decay) as also suggested by [Zhao et al., 2025] (Figure 5, top left panel). We clip gradients to global norm 1.

$$(\eta, \beta) \in [0.004, 0.002, 0.001, 0.0005, 0.0000625, 0.00025, 0.000125]$$
$$\times [0.975, 0.95, 0.9]$$

    – *24 full runs* (Figure 18).

# B Complementary Experimental Results

The results in this section complement the discussion in §3. We organize them in 5 subsections, and report all technical details in §A.

- §B.1 outlines all hyperparameter tuning curves for the setting in Table 1 for SGD (with/without clipping and with/without weight decay) – Figure 8 and 9, RMSprop without momentum – Figure 10, and momentum on top of SignSGD – Figure 11.
- §B.3 validates that $\beta_1 = \beta_2$ is a strong-performing option for Adam at a shorter sequence length. Here, we also show that Signum performance is still suboptimal (cf. Figure 2).
- §B.4 validates that $\beta_1 = \beta_2$ is a strong-performing option for Adam across different batch-sizes. This data, comprising training 500 models, is summarized in Figure 3.
- §B.5 reproduces the Signum-Adam gap on Fineweb [Penedo et al., 2024]. Compared to Figure 2 and the settings above, *here we compare at zero weight decay to eliminate this additional confounder*.
- §B.6 confirms on the validity of our findings when ablating on nuances of Signum and Adam such as initialization and bias correction. These findings complement §3.3.

## B.1 Tuning for Table 1

**Setup Summary.** 160 M parameters LM on SlimPajama, trained for 3.2 B tokens at a batchsize of $256 \times 2048$ sequence length.

**Comment.** Our objective here is to tune to best, despite the combinatorially exploding number of options, our methods in Table 1. Details regarding our hyperparameters grid and model configurations are reported in §A. We remind that tuning for Signum and Adam is presented directly in the main paper as Figure 2. **All figures below show optimal tuning jointly in learning rate and momentum space**. While tuning for RMSprop and momentum on SignSGD is straightforward, SGD requires more attention: we found that removing weight decay was always beneficial when global norm clipping the raw gradient, hence we adopt this option also for the non-clipped variant, and for the variant that includes an additional coordinate clipping step after applying momentum. We believe this is due to the decoupled nature of weight decay, combined with the high learning rates required for good performance in SGD.

**Finalizing Table 1.** After careful tuning, we select for each method the best configuration (given by figures below) and run two additional seeds to report final results with 2-sigma confidence bars.

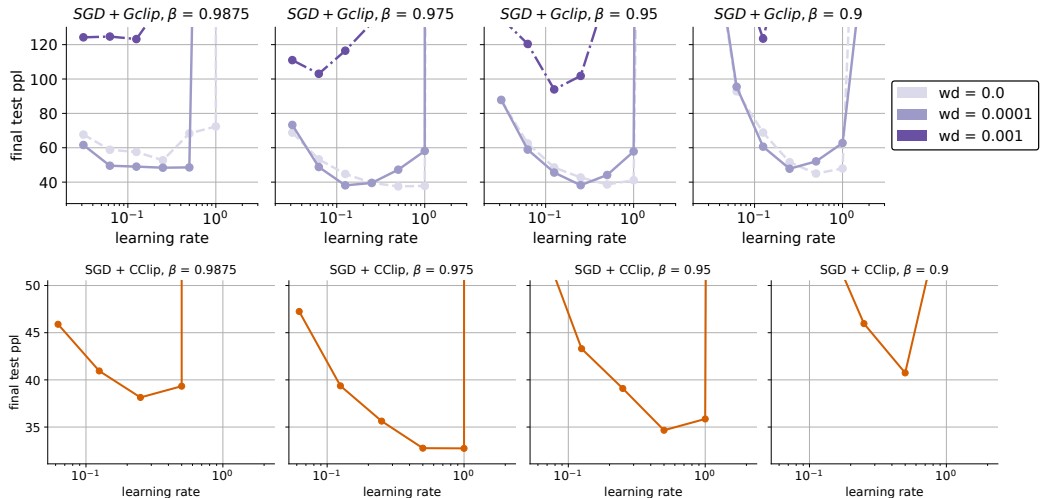

Figure 8: **(top) SGD with global norm clipping**. *We found it beneficial to remove weight decay: the best setting achieves 37.53 ppl, while a slightly larger wd leads to 38.11. a weights decay of 0.001 is too large and yields 93.7 best validation perplexity.* **(bottom) SGD with global norm clipping on raw gradients, followed by coordinate clipping on momentum.** *We remove weight decay as suggested by the top plot. We observe an improvement of 5 perplexity points.*

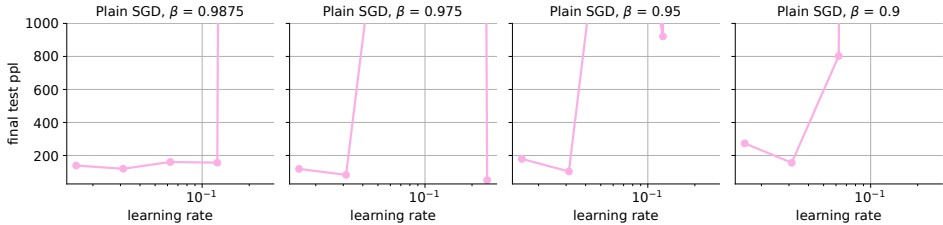

Figure 9: **SGD without coordinate-wise clipping** *at zero weight decay (as suggested by Figure 8 ).*

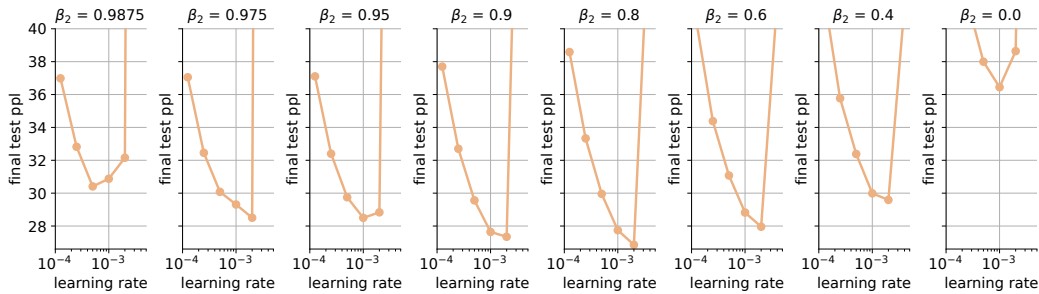

Figure 10: **RMSprop with decoupled weight decay** *0.1. Implemented with Pytorch AdamW setting $\beta_1 = 0$.*

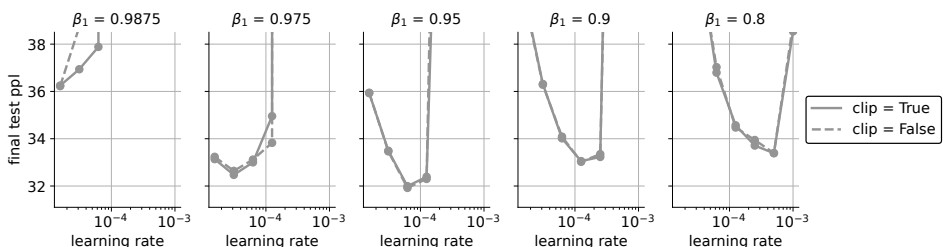

Figure 11: **Momentum on SignSGD** with decoupled weight decay. We implement this just for completeness to show that it is performing worse than `Signum`. Clipping has mathematically no effect (we did not notice at first, so we show the result anyways).

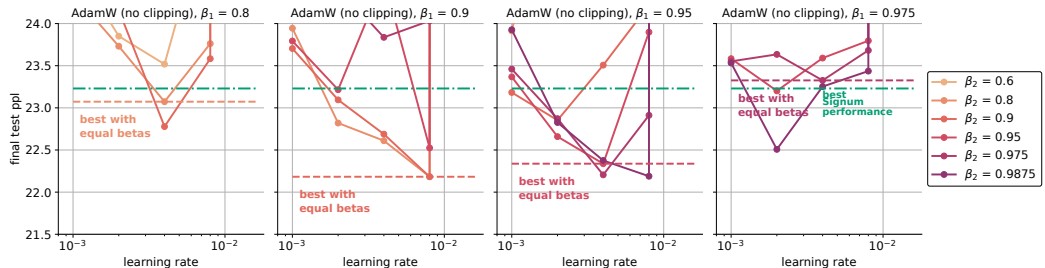

Figure 12: ***AdamW without global norm clipping on gradients*** *with decoupled weight decay. Compared to Figure 2, here we do not clip gradients as a first preprocessing step. Performance is slightly worse, and results are noisier. The best setting, among the ones we tried, is $\beta_1 = \beta_2 = 0.9$. Note, however, that for large/small $\beta_1$s, we observe that some specific configuration with high $\beta_2$ can be beneficial (while still suboptimal if $\beta_1 = \beta_2$ is tuned). In practice, best performance can also be achieved in this setting by merely tuning $\beta_1 = \beta_2 = \beta$, resulting in drastic hyperparameter grid size reduction.*

## B.2 Effect of More Training Tokens in Figure 2

We run part of the experiments in Figure 2 at twice the token budget. Results are conceptually very similar, and show that, on top of $\beta_1 = \beta_2$ being a performance choice for AdamW, that there exists a strong correlation between $\beta$ values (see Fig. 3).

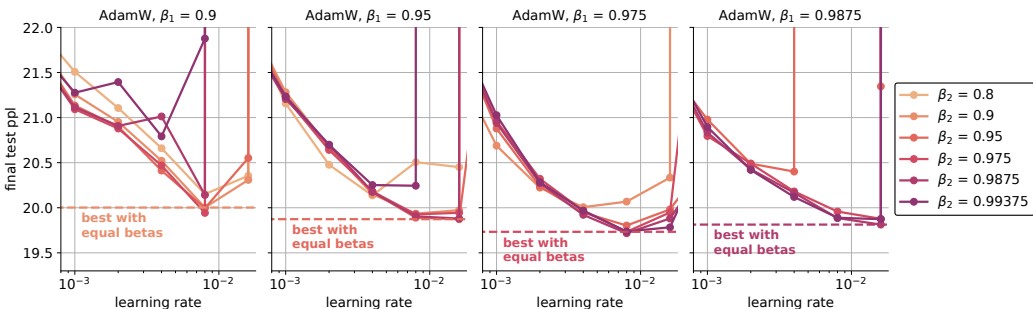

Figure 13: ***AdamW***, *same setting as Figure 2, but trained for twice the number of tokens.*

## B.3  Effect of Shorter Sequence Length in Figure 2

We run part of the experiments in Figure 2 at a lower sequence length (512), for a batch size of 256 sequences (as Figure 2). The model here still sees 3.2B tokens (compute optimal), but number of effective optimizer steps is 4 times bigger compared to the 2048 sequence length setting. While we still observe a sizeable gap between `Signum` and `Adam`, we note that this is smaller compared to Figure 2, as noted also by Zhao et al. [2025] in a similar setting.

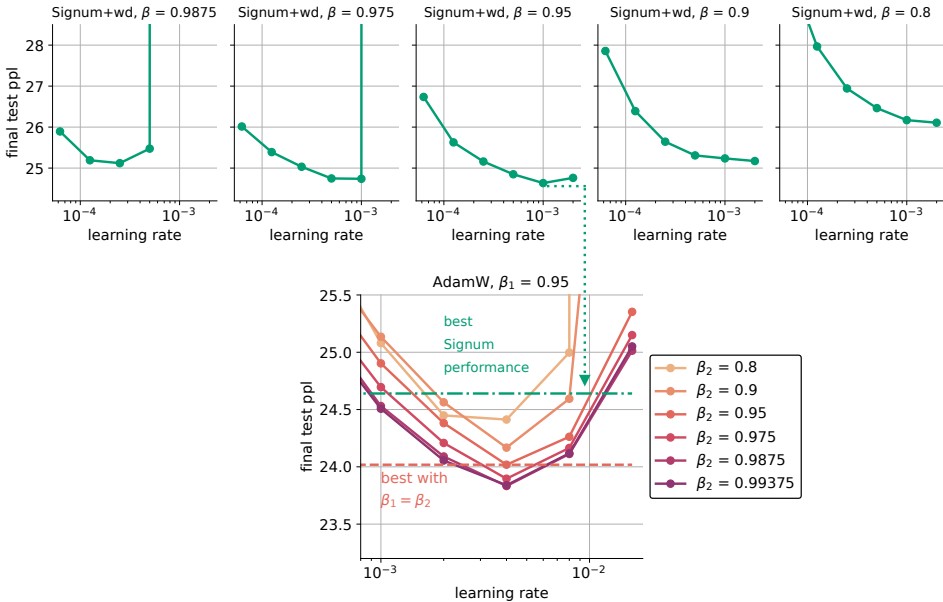

Figure 14: *AdamW vs Signum, same setting as Figure 2, but at a smaller sequence length (512).*

## B.4   Batch size ablation for Figure 2

We run part of the experiments in Figure 2 at a lower and higher batch size. All other details remain the same and are summarized in §A – except for the number of steps performed: due to limitations in our resources, we chose here to train models for 2.5B tokens – i.e. a slightly undertrained setting (optimal would be 3.2B). In line with [Malladi et al., 2022, Compagnoni et al., 2025] we consider half-steps when tuning. All experiments use a weight decay of 0.1.

Despite some imperfections and noise in performance, we notice that $\beta_1 = \beta_2$ is a strong choice even at different batch sizes, our Takeaway 2.

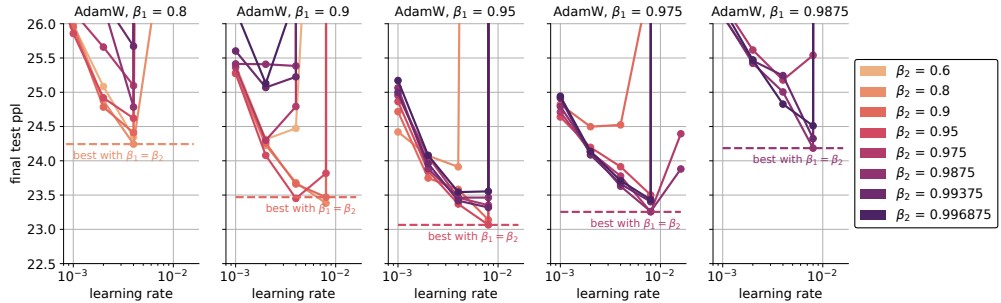

Figure 15: *Adam, batch size 256 trained for 2.5B tokens. Other settings are same setting as Figure 2.*

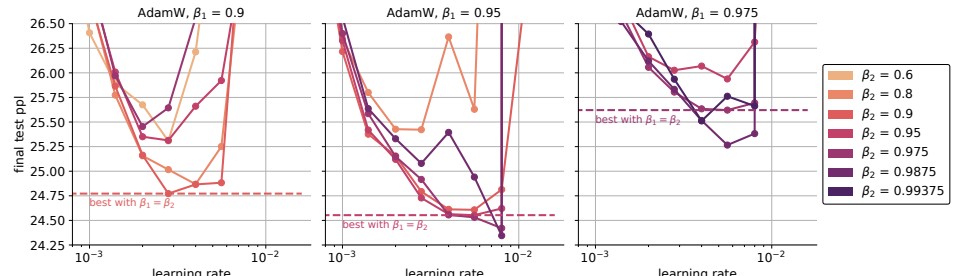

Figure 16: *Adam, batch size 512 trained for 2.5B tokens. Other settings are same setting as Figure 2.*

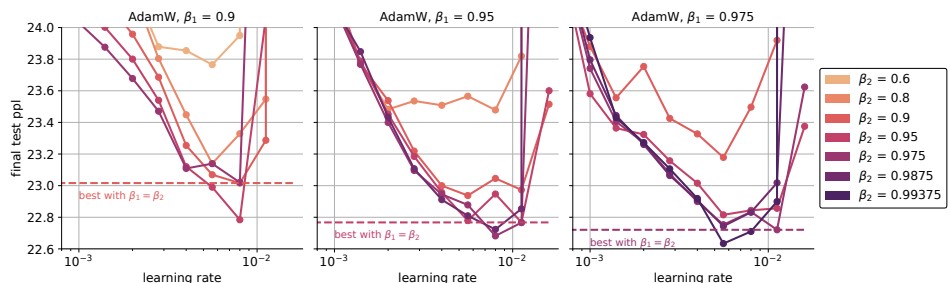

Figure 17: *Adam, batch size 128 trained for 2.5B tokens. Other settings are same setting as Figure 2.*

### B.5 Figure 2 on Fineweb (no weight decay)

Finally, we evaluate our findings – both strong performance of equal $\beta$s in `Adam` and substantial gap with `Signum` on a different dataset (Fineweb [Penedo et al., 2024]). All other experiments in this paper are performed on SlimPajama. To add an additional axis of variation compared to previously presented settings, we here remove weight decay from all methods.

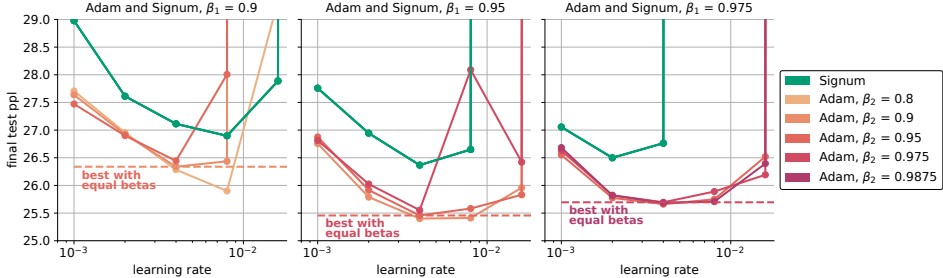

Figure 18: *`Adam` and `Signum` (no weight decay) on Fineweb. Other settings are same as Figure 2.* For visualization purposes, here **we rescaled the visualized learning rate** of `Signum` by a factor $\sim 10$.

### B.6 Effect of Bias Correction and Zero Initialization on `Adam`

The findings below complement our discussion in §3.3.

Table 3: *ZI denotes Zero init of EMA parameters, GI denotes init of EMA parameters to the measurement at first iteration, BC denotes Bias Correction.* ***Not*** *doing ZI means we initialize $m$ and $v$ at $g_0$ and $g_0^2$ respectively. Default for* `Adam` *is ZI and BC. Default for Signum+WD is less clear. We found that initialization does not affect much performance in Signum, yet it does in* `Adam`*. Performing bias correction is not as important as initialization in* `Adam`*. All other parameters in this ablation are fixed to the optimal ones found in default settings for BC and ZI.*

|  | Adam (+ZI+BC) | Adam (+ZI-BC) | Adam (+GI-BC) | Signum (+GI) | Signum (+ZI) |
|---|---|---|---|---|---|
| Val ppl | $21.86\pm 0.21$ | $21.89\pm 0.16$ | $22.58\pm 0.35$ | $23.23\pm 0.16$ | $23.30\pm 0.25$ |

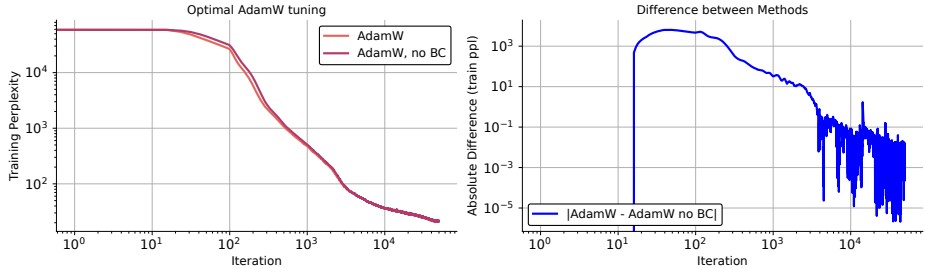

Figure 19: *Effect of eliminating bias correction in* `Adam`*. The difference between variants vanishes as iterations progress. Plotted is the average dynamics over 3 random seeds.*

## C    Missing proofs and derivations

### C.1    Proof of Proposition 1

**Proposition 1.** *Let $m_k = \mathtt{EMA}_\beta[g_k]$. Then the update* (2) *admits the equivalent representation:*

$$d_k = \frac{m_k}{\sqrt{m_k^2 + \beta\, \mathtt{EMA}_\beta[(m_{k-1} - g_k)^2]}}. \tag{3}$$

*Proof of Proposition 1 .* For this proof we will use the abbreviation

$$v_k := \mathtt{EMA}_\beta[g_k^2].$$

With this abbreviation the $\mathtt{Adam}$ update (2) can be written as

$$d_k = \frac{m_k}{\sqrt{v_k}} = \frac{m_k}{\sqrt{m_k^2 + v_k - m_k^2}}.$$

Next we will show that $v_k - m_k^2 = \beta\mathtt{EMA}_\beta[(m_{k-1} - g_k)^2]$. Indeed by expanding the update of $v_{k+1}$ and $m_{k+1}$ we have that

$$
\begin{aligned}
v_{k+1} - m_{k+1}^2 &= \beta v_k + (1-\beta)g_{k+1}^2 - (\beta m_k + (1-\beta)g_{k+1})^2 \\
&= \beta v_k + (1-\beta)g_{k+1}^2 - (\beta^2 m_k^2 + (1-\beta)^2 g_{k+1}^2 + 2\beta(1-\beta)g_{k+1}m_k) \\
&= \beta v_k - \beta^2 m_k^2 + (1-\beta)\beta g_{k+1}^2 - 2\beta(1-\beta)g_{k+1}m_k \\
&= \beta v_k - \beta m_k^2 + \beta m_k^2 - \beta^2 m_k^2 + (1-\beta)\beta g_{k+1}^2 - 2\beta(1-\beta)g_{k+1}m_k \\
&= \beta(v_k - m_k^2) + \beta(1-\beta)m_k^2 + \beta(1-\beta)g_{k+1}^2 - 2\beta(1-\beta)g_{k+1}m_k \\
&= \beta(v_k - m_k^2) + \beta(1-\beta)(m_k - g_{k+1})^2.
\end{aligned}
$$

By setting $\delta_k = v_k - m_k^2$ we have that

$$\delta_{k+1} = \beta\delta_k + \beta(1-\beta)(m_k - g_{k+1})^2 = \beta\mathtt{EMA}_\beta[(m_{k-1} - g_k)^2]$$

where we used the definition of the EMA recurrence in (1).

$\square$

### C.2    Generalization of Proposition 1 – Necessity of equal betas for variance interpretation

**Proposition 2.** $\mathtt{Adam}$ *with hyperparameters $\beta_1, \beta_2 \in (0,1)$ has update of form*

$$\frac{m_k}{\sqrt{m_k^2 + \gamma \mathtt{EMA}_\tau[(am_{k-1} - bg_k)^2]}},$$

*for some $a, b, \gamma \in \mathbb{R}$ and $\tau \in (0,1)$ if an only if $\beta_1 = \beta_2$.*

*Proof of Proposition 2.* Let us expand the expression.

$$
\begin{aligned}
v_{k+1} - m_{k+1}^2 &= \beta_2 v_k + (1-\beta_2)g_{k+1}^2 - (\beta_1 m_k + (1-\beta_1)g_{k+1})^2 \\
&= \beta_2 v_k + (1-\beta_2)g_{k+1}^2 - [\beta_1^2 m_k^2 + (1-\beta_1)^2 g_{k+1}^2 + 2\beta_1(1-\beta_1)m_k g_{k+1}] \\
&= \beta_2 v_k - \beta_1^2 m_k^2 + [(1-\beta_2) - (1-\beta_1)^2]g_{k+1}^2 - 2\beta_1(1-\beta_1)m_k g_{k+1}
\end{aligned}
$$

**The case of equal betas.**    Notice that if $\beta_1 = \beta_2 = \beta$, then

$$(1-\beta) - (1-\beta)^2 = 1 - \beta - (1 + \beta^2 - 2\beta) = 1 - \beta - 1 - \beta^2 + 2\beta = \beta(1-\beta),$$

and so the expression gets simplified:

$$v_{k+1} - m_{k+1}^2 = \beta v_k - \beta^2 m_k^2 + \beta(1-\beta)[g_{k+1}^2 - 2m_k g_{k+1}]$$

Now add and subtract $\beta m_k^2$, to get

$$v_{k+1} - m_{k+1}^2 = \beta(v_k - m_{k+1}^2) + \beta(1-\beta)[m_k^2 + g_{k+1}^2 - 2m_k g_{k+1}].$$

where the last term is the perfect square $(m_k - g_{k+1})^2$.

**The general setting.** One might hope for the "stars aligning" into a perfect square also in the general setting. For this to happen, we need to require that the term

$$[(1 - \beta_2) - (1 - \beta_1)^2]g_{k+1}^2 - 2\beta_1(1 - \beta_1)m_k g_{k+1}$$

allows for such a simplification to happen. That is, assume to start from

$$(am_k - bg_{k+1})^2 = a^2 m_k - 2abm_k g_{k+1} + b^2 g_{k+1}^2.$$

we need

$$b^2 = (1 - \beta_2) - (1 - \beta_1)^2, \quad ab = \beta_1(1 - \beta_1).$$

so

$$a = \frac{\beta_1(1 - \beta_1)}{\sqrt{(1 - \beta_2) - (1 - \beta_1)^2}}.$$

Therefore:

$$\left( \frac{\beta_1(1 - \beta_1)}{\sqrt{(1 - \beta_2) - (1 - \beta_1)^2}} m_k - \sqrt{(1 - \beta_2) - (1 - \beta_1)^2} g_{k+1} \right)^2$$

$$= \frac{\beta_1^2(1 - \beta_1)^2}{(1 - \beta_2) - (1 - \beta_1)^2} m_k^2 + [(1 - \beta_2) - (1 - \beta_1)^2]g_{k+1}^2 - 2\beta_1(1 - \beta_1)m_k g_{k+1}$$

Therefore, in the general setting, we can write

$$v_{k+1} - m_{k+1}^2 = \beta_2 v_k - \left( \beta_1^2 + \frac{\beta_1^2(1 - \beta_1)^2}{(1 - \beta_2) - (1 - \beta_1)^2} \right) m_k^2 +$$

$$+ \left( \frac{\beta_1(1 - \beta_1)}{\sqrt{(1 - \beta_2) - (1 - \beta_1)^2}} m_k - \sqrt{(1 - \beta_2) - (1 - \beta_1)^2} g_{k+1} \right)^2$$

Massaging a bit, we get

$$v_{k+1} - m_{k+1}^2 = \beta_2 v_k - \frac{\beta_1^2(1 - \beta_2)}{(1 - \beta_2) - (1 - \beta_1)^2} m_k^2 +$$

$$+ \left( \frac{\beta_1(1 - \beta_1)}{\sqrt{(1 - \beta_2) - (1 - \beta_1)^2}} m_k - \sqrt{(1 - \beta_2) - (1 - \beta_1)^2} g_{k+1} \right)^2$$

which implies

$$v_{k+1} - m_{k+1}^2 = \beta_2 \left( v_k - \frac{\beta_1^2(1 - \beta_2)}{\beta_2(1 - \beta_2) - \beta_2(1 - \beta_1)^2} m_k^2 \right) +$$

$$+ \left( \frac{\beta_1(1 - \beta_1)}{\sqrt{(1 - \beta_2) - (1 - \beta_1)^2}} m_k - \sqrt{(1 - \beta_2) - (1 - \beta_1)^2} g_{k+1} \right)^2.$$

Therefore, the formula holds true if and only if

$$\frac{\beta_1^2(1 - \beta_2)}{\beta_2(1 - \beta_2) - \beta_2(1 - \beta_1)^2} = 1.$$

That is, if and only if

$$\beta_1^2(1 - \beta_2) = \beta_2(1 - \beta_2) - \beta_2(1 - \beta_1)^2.$$

The condition simplifies, as it reads:

$$\beta_1^2 - \beta_1^2\beta_2 = \beta_2 - \beta_2^2 - \beta_2 - \beta_2\beta_1^2 + 2\beta_1\beta_2.$$

which simplified is

$$\beta_1^2 + \beta_2^2 - 2\beta_1\beta_2 = 0.$$

i.e.

$$(\beta_1 - \beta_2)^2 = 0 \quad \Longleftrightarrow \quad \beta_1 = \beta_2.$$

$\square$

## C.3 Proof of Theorem 4.1

**Theorem 4.1.** Let $\beta = \frac{1}{1+\lambda}$. Then the solution to the optimization problem (4) is given by

$$m_{k+1} = \beta m_k + (1-\beta)g_{k+1} = \mathtt{EMA}_\beta[g_{k+1}], \tag{7}$$

$$\sigma_{k+1}^2 = \beta\sigma_k^2 + \beta(1-\beta)(m_k - g_{k+1})^2 = \beta\,\mathtt{EMA}_\beta\left[(m_k - g_{k+1})^2\right]. \tag{8}$$

*Proof.* Recall that

$$-\log p(g_{k+1} \mid m, \sigma^2) = \frac{1}{2}\log\sigma^2 + \frac{1}{2\sigma^2}(g_{k+1} - m)^2,$$

$$\mathrm{KL}\left(\mathcal{N}(m_k, \sigma_k^2) \,\|\, \mathcal{N}(m, \sigma^2)\right) = \frac{1}{2}\left[\frac{\sigma_k^2}{\sigma^2} + \frac{(m_k - m)^2}{\sigma^2} - 1 - \log\left(\frac{\sigma_k^2}{\sigma^2}\right)\right].$$

Therefore

$$F(m, \sigma^2) = -\log p(g_{k+1} \mid m, \sigma^2) + \frac{1}{\lambda}\mathrm{KL}\left(\mathcal{N}(m_k, \sigma_k^2) \,\|\, \mathcal{N}(m, \sigma^2)\right)$$

$$= \frac{1}{2}\log\sigma^2 + \frac{1}{2\sigma^2}(g_{k+1} - m)^2 + \frac{1}{2\lambda}\left[\frac{\sigma_k^2}{\sigma^2} + \frac{(m_k - m)^2}{\sigma^2} - 1 - \log\left(\frac{\sigma_k^2}{\sigma^2}\right)\right]$$

Since we are not optimizing for $\sigma_k^2$, we can replace $-\log\left(\frac{\sigma_k^2}{\sigma^2}\right) = \log(\sigma^2)$ and drop constants, gives the following objective function

$$\min_{m,\sigma^2 \geq 0} F(m, \sigma^2) = \frac{1}{2}\frac{1+\lambda}{\lambda}\log(\sigma^2) + \frac{1}{2\sigma^2}\left[(g - m)^2 + \frac{1}{\lambda}\left(\sigma_k^2 + (m_k - m)^2\right)\right] + \mathrm{const.}$$

**Stationarity in $m$:** Differentiating in $m$ and setting to zero gives

$$\frac{\partial F}{\partial m} = -\frac{1}{\sigma^2}(g - m) - \frac{1}{\lambda\sigma^2}(m_k - m) = 0.$$

Multiplying by $\lambda\sigma^2$, we get:

$$-\lambda(g - m) - (m_k - m) = 0 \quad \Rightarrow \quad m = \frac{\lambda g + m_k}{1 + \lambda}. \tag{13}$$

**Stationarity in $\sigma^2$:** Differentiating in $\sigma^2$ and setting to zero gives

$$\frac{\partial F}{\partial\sigma^2} = \frac{1}{2}\frac{1+\lambda}{\lambda}\cdot\frac{1}{\sigma^2} - \frac{1}{2\sigma^4}\left[(g - m)^2 + \frac{1}{\lambda}\left(\sigma_k^2 + (m_k - m)^2\right)\right] = 0.$$

Multiplying both sides by $2\sigma^4$, and re-arranging gives:

$$\frac{1+\lambda}{\lambda}\sigma^2 = (g - m)^2 + \frac{1}{\lambda}\left(\sigma_k^2 + (m_k - m)^2\right).$$

Multiplying through by $\frac{\lambda}{1+\lambda}$ gives

$$\sigma^2 = \frac{\lambda(g - m)^2 + \left[\sigma_k^2 + (m_k - m)^2\right]}{1 + \lambda}. \tag{14}$$

Now using $m = \frac{\lambda g + m_k}{1+\lambda}$ from (13) we have that

$$g - m = \frac{g - m_k}{1 + \lambda}, \quad m_k - m = \frac{\lambda(m_k - g)}{1 + \lambda}.$$

Therefore:

$$(g - m)^2 = \frac{(g - m_k)^2}{(1 + \lambda)^2}, \quad (m_k - m)^2 = \frac{\lambda^2(g - m_k)^2}{(1 + \lambda)^2}.$$

Using the above in the expression for $\sigma^2$ in (14), we get:

$$\sigma^2 = \frac{\lambda(g - m_k)^2}{(1 + \lambda)^2} + \frac{\sigma_k^2}{1 + \lambda}.$$

This, together with (13) gives the final solution

$$\boxed{m_{k+1} = \frac{m_k + \lambda g}{1 + \lambda}} \quad \text{and} \quad \boxed{\sigma_{k+1}^2 = \frac{\sigma_k^2}{1 + \lambda} + \frac{\lambda(g - m_k)^2}{(1 + \lambda)^2}}.$$

If we use the standard momentum parameterization, which corresponds to $\beta = \frac{1}{1+\lambda}$ we arrive at the stated results (7) and (8) of the theorem. □

### C.4 Performance of generalized Adam reformulation

As described in §4.2, we here consider performance of the update direction:

$$d_k = \frac{m_k}{\sqrt{m_k^2 + \gamma \, \texttt{EMA}_\tau[(m_{k-1} - g_k)^2]}} \tag{AdaVar}$$

This reduces to `Adam` with equal betas as soon as $\beta = \gamma = \tau$ but cannot be written as an `Adam` update as soon as $\beta \neq \gamma$ or $\gamma \neq \tau$ (see proof in §C.2). Further, our theory in §4 shows that $\beta = \gamma = \tau$ is the only theoretically grounded choice for a precise online variational inference interpretation, also in this setting, i.e. when considering $\sigma_k^2 = \gamma \, \texttt{EMA}_\tau[(m_{k-1} - g_k)^2]$. We wonder if this insight correlates with optimal performance.

As one can see in Figure 20, we found that setting $\beta = \tau = \gamma$ leads to near optimal performance in all settings.

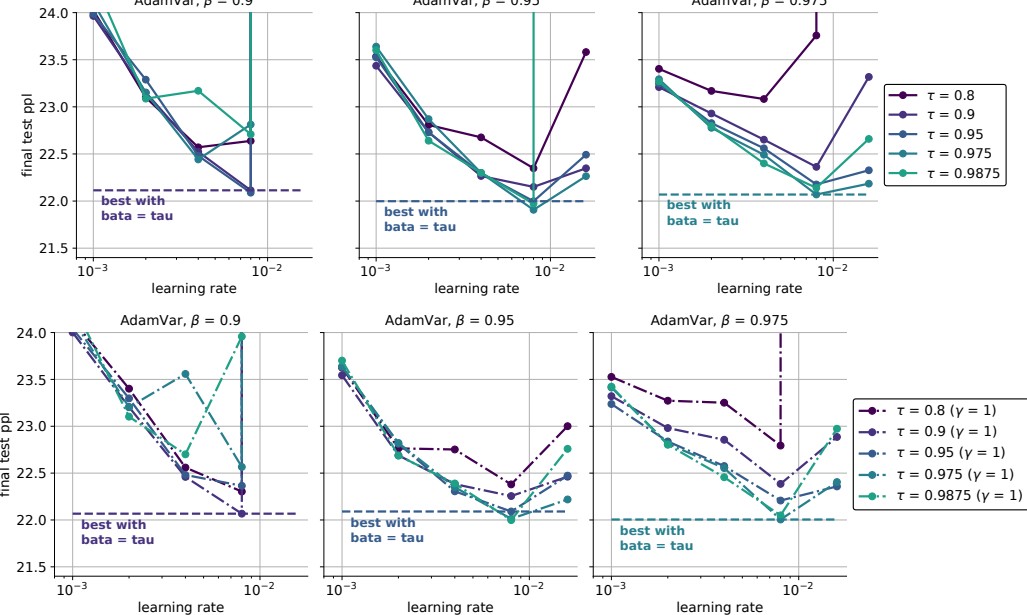

Figure 20: *Performance of AdaVar **aligns with our theoretical insights**. Setup for these experiments is exactly the same as for Figure 2.*

# D  Toy Quadratic Example

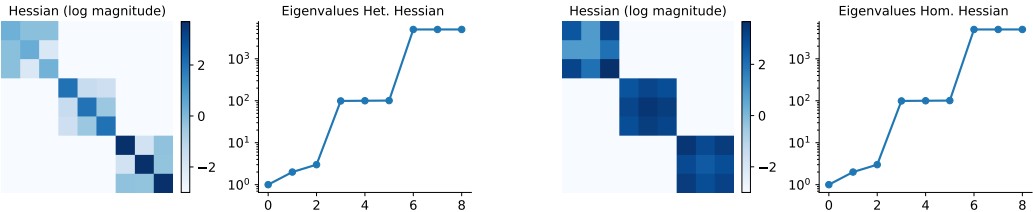

Figure 21: *(left) Heterogeneous and (right) Homogeneous Hessian considered in §5.*

Our setup here is inspired directly from the results and discussions in Zhang et al. [2024a]. Specifically, we consider the loss

$$L(w) = \frac{1}{2} w^\top H w$$

where we construct the Homogeneous and Heterogeneous Hessians using the following procedure:

- We fix the eigenvalues, equal in both cases, to

$$\operatorname{eig}(H_{\text{hom}}) = \operatorname{eig}(H_{\text{het}}) = \{1, 2, 3, 99, 100, 101, 4998, 4999, 5000\}.$$

- We choose both Hessians to be block-diagonal, with blocks of size $3 \times 3$. The homogeneous Hessian has eigenvalues of different magnitude in each block, while the Heterogeneous keeps similar magnitudes in each block.

```
H_details_het = [[1,2,3],[99,100,101],[4998,4999,5000]]
H_details_hom = [[1,99,4998],[2,100,4999],[3,101,5000]]
```

- For each block, we apply a random rotation to the diagonal matrix of eigenvalues, specific to each block. Each rotation is sampled from the Haar measure by decomposition of a random $3 \times 3$ positive semidefinite matrix $AA^\top$, where $A \in \mathbb{R}^{3 \times 3}$ has i.i.d. Gaussian entries.

The result is shown in Figure 21.

Next, to introduce stochasticity in this setting, we simply take the square root of the Hessian to define a $9 \times 9$ design matrix $X$

$$H = X^\top X, \qquad X = H^{\frac{1}{2}}$$

and subsample a number (the batchsize) of rows of $X$ at each iteration.

# E   Signal Processing Perspective

In this last section, we examine `Adam` through a signal processing lens, to get qualitative insights into its distinction with `Signum` and other `SignSGD` with momentum variants. Setting $\beta_1 = \beta_2 = \beta$, we can write the `Adam` update, without bias correction (see §B.6) as simply

$$d_k = \left( \sqrt{\text{EMA}_\beta[g_k^2]} + \epsilon \right)^{-1} \text{EMA}_\beta[g_k]$$

where $(g_k)_k$ is the gradient signal. One might wonder if this special case allows for a simpler graphical interpretatoin of `Adam`. To do this, **we consider here fixing the gradient signal, and see how different methods process this signal.**

**Graphical intuition.**   We denote by $d_k$ the update of `Adam` once it sees a gradient signal $(g_i)_{i \leq k}$:

and plot its dynamics as a function of a synthetic one-dimensional gradient in Figure 22.

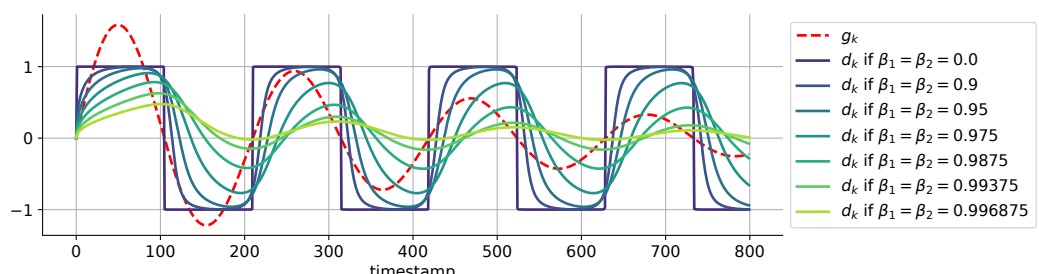

Figure 22: Filtering effect for same $\beta_1 = \beta_2$.

In the example of Figure 22, we chose the synthetic gradient signal

$$g_k = 1.8 \sin(0.03k) \exp(-0.0025k)$$

this is a damped periodic signal plotted in red. Note that this is pure filtering, there is no loss or learning process. We note the following:

1. $\beta_1 = \beta_2 = 0$ is obviously just $\text{sign}(g_k)$. This is plotted for comparison.
2. For any $\beta_1 = \beta_2 \neq 0$, $d_k$ is bounded by 1 in magnitude. It's dynamics however, for e.g. $\beta_1 = \beta_2 > 0$ is smooth and follows more closely the gradient, while being bounded. It is somehow a rescaled version. More on this later.
3. Very interestingly, $d_k$ is blind to the decay term $\exp(-0.0025k)$, the output is perfectly periodic for every $\beta_1 = \beta_2$.

Towards proceeding, note that $d_k$ **cannot be reduced to momentum on the sign or sign on the momentum(Signum)**: both variants actually destroy the signal shape, while $d_k$ maintains the shape of the original signal and has clear invariance properties. The behavior of signSGD with momentum (2 variants) is shown in Figure 23: as one can see, the behavior is drastically different from $d_k$ in Figure 22, an enlargement is shown in Figure 24.

We now try to formalize some of the properties we observe.

**Properties.**   `Adam` can be seen as a very special operator $T$ on gradient sequences $(g_k)_{k=0}^\infty \in \mathcal{G} \subseteq \ell_\infty$ (with normed vector space structure and notation). We can identify four distinctive properties. $T : (g_k)_{k=0}^\infty \to (d_k)_{k=0}^\infty$.

1. It is **causal**.
2. It is **invariant to positive scaling**: $T(\alpha \cdot g) = T(g)$, for any $\alpha > 0$.
3. It is **odd**: $T(-g) = -T(g)$.
4. It has **bounded** infinity norm: $\|T(g)\|_\infty \leq 1$ for all $g \in \ell_\infty$.

5. **Density**: For any $b \in [-1, 1]$ and any arbitrary $k > 0$, there exists $(g_k)_{k=0}^{\infty}$ such that $d_k = b$.

We are amazed by these rich set of properties, thickening our interest in better understanding the properties of `Adam` mollification, which we study in §4.

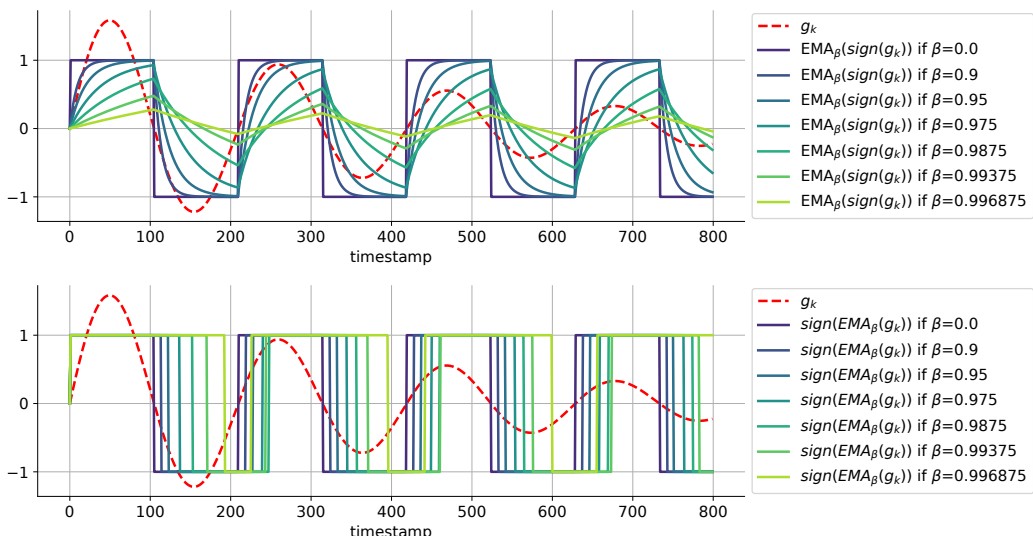

Figure 23: Filtering induced by signSGD with momentum (2 variants, the one below is `Signum`). Compare with Figure 22.

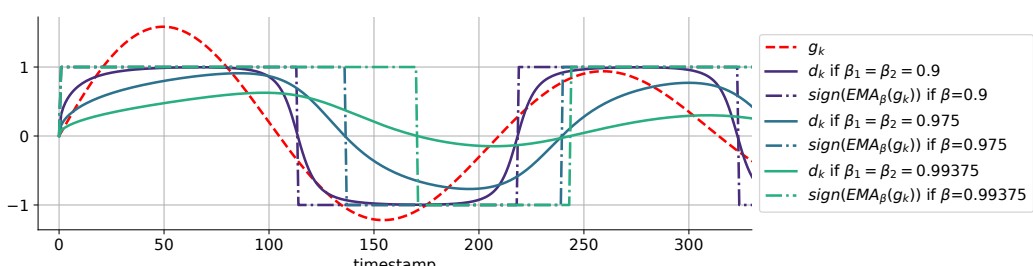

Figure 24: `Adam`-like filtering compared to sign of EMA (`Signum`), detail.

We hope this investigation ispires future effors in understanding these intriguing phenomena and properties. We conclude the paper with a quote, stolen from the Bernt Øksendal masterpiece book on SDEs:

*We have not succeeded in answering all our problems.*
*The answers we have found only serve to raise a whole set*
*of new questions. In some ways we feel we are as confused*
*as ever, but we believe we are confused on a higher level*
*and about more important things.*

Posted outside the mathematics reading room –Tromsø University

