# OpenReview forum: "In Search of Adam’s Secret Sauce"
_NeurIPS.cc/2025/Conference — NeurIPS 2025 oral_

### Official Review · Reviewer_Gwdg · 2025-07-01

**Clarity:** 4
**Significance:** 4
**Originality:** 4
**Rating:** 5
**Confidence:** 4

**Summary:**

This paper firstly provides a comprehensive empirical study of the Adam optimizer on pertaining language models with extensive model types and hyperparameter settings. Through these experiments, the authors found out that the optimal beta parameter setting in their experiments is $\beta_1=\beta_2$, in which case they theoretically derived a new interpretation of the Adam updates as estimating the mean and variance of the stochastic gradients through variational inference. Under this framework, they interpret the Adam update as direction search within an adaptive trust region based on the ratio between mean and variance of the gradients (signal-to-noise ratio), contrasting with Signum (SignSGD with momentum), whose trust region is fixed in the lens of this framework. In addition, a landscape heterogeneity interpretation of such advantage is provided and validated with a constructed quadratic example.

**Questions:**

1. The empirical and theoretical analysis is done by contrasting with SignSGD and those interpolating between SGD and Adam, in particular for the theoretical analysis. The reviewer is wondering do the authors have additional comments on Adam if in contrast with other recent optimizers like Shanpoo [Gupta et al., 2018, Vyas et al. 2025], Lion [Chen et al. 2023], which are mentioned in the related work but not quite discussed. For Lion in particular, as a generalized version of SignSGD, does its improvement on SignSGD, to some extent, close this gap to Adam, or seem to be orthogonal efforts?

**Ethical Concerns:**

["NO or VERY MINOR ethics concerns only"]

**Final Justification:**

The authors addressed my questions in the rebuttal, and the paper is overall insightful. Therefore, I'm keeping my score and supporting the acceptance of this paper.

**Limitations:**

yes

**Quality:**

4

**Strengths And Weaknesses:**

**Strengths:**
1. The empirical study of Adam is very comprehensive on training language models.
2. The finding of $\beta_1=\beta_2$ as the optimal Adam hyperparameter setting is of practical guidance to the broader ML community.
3. The theoretical interpretation of Adam from the variational inference perspective is novel to the best of the reviewer's knowledge, and the reviewer finds the adaptive trust-region perspective of Adam when contrasting with SignSGD very insightful.

**Weaknesses:**
1. Even though many hyperparameter combinations are tried out, the space grows exponentially, and the grid included in the paper could still be limited, looking at one parameter, e.g., $\eta$, alone, as the authors acknowledge this limitation in the conclusion
2. The experiments are only conducted for training language models, excluding other tasks or data like images, which could limit the generality of the findings.
3. The variational inference interpretation only holds when $\beta_1=\beta_2$, while other values in different settings may still show superior, sometimes optimal performance, which could limit the generality of this theory, or leave a gap between theory and practice.
4. Bias correction seems not included in the theoretical analysis. Though the emperical results do not seem to be affected, the bias correction of Adam is usually regarded as an important theoretical component.

---

> ### Author Rebuttal · Authors · 2025-07-29
>
> We thank the reviewer for their time, their insightful comments, and for finding our work comprehensive, practical, and novel!
>
> We answer your questions below:
>
> ---
>
> **1) Other domains**
>
> The reviewer is absolutely correct: this question is of extreme interest. We limited ourselves to the LM setting, as this discussion already encompasses multiple scenarios (batch sizes, regularization, data, and scale), is compute-intensive and often used to showcase the Adam-SGD gap. We, however, note that a concurrent paper, "Matias D. Cattaneo and Boris Shigida. Tuning Adam(w): Default beta2 may be too large, 2025", which we cite in line 332, provides ablations on small-scale vision domains confirming the default $\beta_2=0.999$ in Adam is too high.
>
> ---
>
> **2) Variational Interpretation for $\beta_1\ne\beta_2$**
>
> Our variational interpretation stems directly from our empirical finding that $\beta_1=\beta_2$ often leads to optimal results (e.g., Figure 2 and Figure 4).
> 1. Let us elaborate better what we mean: while for a fixed arbitrary $\beta_1$ it might indeed be that $\beta_2>\beta_1$ leads to *slightly* better results (maybe due to noise, e.g. Figure 2, $\beta_1=0.9875$), at the optimal $\beta_1$ we have both in Figure 2 and Figure 4 that $\beta_2=\beta_1$ leads to best perplexity. This implies that if one tunes only $\beta_1$ and sets $\beta_2=\beta_1$, then the optimal result can be achieved.
> 2. Our point is that Adam with $\beta_1=\beta_2$ is not only a good method but also a statistically principled method. Our novelty in this regard is to demonstrate that **only** in this setting does Adam admit a strong statistical viewpoint, as we show in Appendix C.2. We will reference this appendix and provide further commentary on it in our manuscript. Thanks for the question!
> 3. To conclude, $\beta_1=\beta_2$ leads to a simplified and statistically principled method, which performs well (= much better than Signum) and often optimally in practice. This is not only interesting but also gives clear guidelines for tuning.
>
> ---
>
> **3) Bias Correction**
>
> We thank the reviewer for this careful question. Our analysis doesn't include bias correction because of the way we initialize the momentum buffers in Eq (1). Bias correction was introduced because the momentum (or EMA) estimate is biased for stationary signals when you initialize momentum at zero. We take care of this issue by instead initializing our momentum estimates at the first sample. This already guarantees an unbiased estimate of the momentum estimate, see Section 2.3 here for a discussion.
>
> In any case, we also show in Section B.5. (referenced at line 221) that bias correction does not affect qualitatively our results. We agree that some works devote much focus on this, but also point out that bias correction can be seen as simply modifying learning rate scheduling (e.g. see equation 5 in https://arxiv.org/pdf/2003.02395), and our statistical interpretation is independent of scheduling.
>
> ---
>
> **4) Other methods, e.g., Shampoo, Lion**
>
> In this paper, we address the question of understanding Adam with *simpler* methods that claim *theoretical* similarity with this optimizer. Both Shampoo and Lion are designed to surpass Adam, not to simplify it or to explain its update rule. For this reason, we believe comparing our methods with these would distract from the message in our paper, which is not devoted to finding the best method, but rather to understanding the Adam optimizer and how to simplify its tuning process.
>
> [1] MoMo: Momentum Models for Adaptive Learning Rates, Fabian Schaipp, Ruben Ohana, Michael Eickenberg, Aaron Defazio, Robert M. Gower, ICLR 2024

---

> > ### Comment · Reviewer_Gwdg · 2025-08-04
> >
> > Thank the authors for their reply. It addresses my questions and I'm keeping my score.

---

### Official Review · Reviewer_rfMp · 2025-07-02

**Clarity:** 3
**Significance:** 3
**Originality:** 2
**Rating:** 5
**Confidence:** 4

**Summary:**

This paper shows, through a large-scale experiment, that the Adam optimiser generally performs better when $\beta_1 = \beta_2$. The authors then propose a simplified version of Adam without bias correction, where  $\beta_1 = \beta_2$ and $\epsilon = 0$. They show that when $\beta_1 = \beta_2$, Adam provides an online estimate of the mean and (centred) variance of the gradient. Further analysis also highlights a close relationship with signum, where the derived version of Adam is shown to be a signum modulated by the squared coefficient of variation (SCV). In other words, the higher the spread/uncertainty in a specific region of the loss landscape, the smaller the update magnitude.

**Questions:**

[1] In Proof C3, l. 530, I was unable to reproduce the given objective function. Instead, when substituting Eqs. 5 and 6 into Eq. 4, I have:
$$\\frac{\\lambda + 1}{2} \\log \\sigma^2 + \\frac{1}{2\\sigma^2}\[ (g_{k+1} - m) + \\lambda(\\sigma^2_k + (m_k - m^2)) \] - \frac{\\lambda}{2} \\log \\sigma_k^2 - \\frac{\\lambda}{2}.$$
I assume the last term is what goes into $const$ but the coefficient for $\\log \\sigma^2$ and $\\sigma^2_k + (m_k - m^2)$ are different and I am not sure what happened to $ \frac{\\lambda}{2} \\log \\sigma_k^2$. Could the authors explain how they arrived at their results? I will be happy to raise my score if I missed a simplification or if there was a typo somewhere that is corrected and does not impact the results of Sec. 4.1.

[2] l. 514, I do not understand how the authors went from $\\beta \\delta_n + \\beta(1- \\beta)(m_k - g_{k+1})^2$ to $\\beta EMA_{\\beta}[(m_{k-1} - g_k)^2]$. If the authors could provide the complete derivation, it would be much appreciated and may increase my score. It would also be great to have an interpretation of the $m_k^2$ factor in the denominator of Eq. 3.

[3] There are several small typos in the maths, which did not impact the results, but should be corrected:

- l. 241 $\\sigma_k$ should be $\\sigma_k^2$ to be consistent with Eq. 8.

- l. 521 shouldn't we have $\\beta(v_k - m_k^2)$ instead of $\\beta(v_k - m_k)$ as we add and subtract $\\beta m_k^2$?

- Before l. 523, when developing the square, we should have $a^2m^2_k$ instead of $a^2m_k$.

- l. 529 not a math typo but an equation which is not referenced: it should be `substituting (5) and (6) into (4)'.

[4] Could we see a comparison of Adam and the update proposed in Eq.9 to assess how much the various assumptions made during the theoretical analysis impact the practical application?

[5] Could the authors explain in what regard their claim that Adam is an online estimate of the mean and variance of the gradient is novel compared to what is stated in the initial paper? I would be happy to raise my score if a small paragraph discussing the differences between the two were added somewhere in the paper, as this may help other readers better understand the novelty of the paper.

**Ethical Concerns:**

["NO or VERY MINOR ethics concerns only"]

**Final Justification:**

The authors have satisfactorily answered my questions, and I am now confident that all the derivations are correct and sufficiently detailed. I have thus raised my score to 5.

**Limitations:**

yes

**Paper Formatting Concerns:**

Most of the figures are too small to be read. A suggestion for Fig.2 and similar figures would be to select only the best LR for each $\\beta_1$, use the x-axis for $\\beta_2$ and a second y-axis for $\\beta_1$. That way, there will be one subfigure for signum and one for Adam instead of 5 for each. For completeness, the full LR details can be included in the appendix, where space limitations are not an issue.

**Quality:**

2

**Strengths And Weaknesses:**

___Quality:___

[+] Most of the derivations and proofs are correct.

[+] The proposed empirical experiment is done on several models and datasets with multiple seeds and an extensive grid search.

[-] I could not reproduce some of the derivations. This is my primary concern as one of the results directly impacts the soundness of Thm 4.1 and the results of Section 4.1. (see questions 1-2)

[-] There are some minor typos in several equations, which did not impact the final results (see question 3)

[-] The impact of the different assumptions made for the theoretical analysis is not assessed experimentally (see question 4).

___Clarity:___

[+] The paper is well-written and easy to follow.

[-] Most of the figures are too small and not black and white printer-friendly. (see comments on formatting concerns)

___Significance:___

[+] The authors provide a large-scale experiment and a theoretical analysis, which are, to my knowledge, new.

[+] This may be used by others to propose a new Adam-based optimiser

[+] The results highlight the importance of having optimisers robust to a heterogeneous landscape, especially in NLP. This may help others to design more efficient optimisers.

___Originality:___

[+] The paper provides an interesting analysis of Adam, which is, to my knowledge, novel, but I am not an expert in the field, and I may not be aware of closely related work.

[-] In the original Adam paper, section 2, the authors said that `the moving averages themselves are estimates of the first moment (the mean) and the second moment (the uncentred variance) of the gradient'. As one of the central claims of the paper is to show that Adam is an online estimate of the mean and variance of the gradient, it may not be very novel. (see question 5)

---

> ### Author Rebuttal · Authors · 2025-07-29
>
> Thank you for the time and effort you have put into reading our paper.
>
> Below we answer all of your questions, including your main question on verifying the proof of Theorem 4.1 (*which resulted from a small type-O*). See below. We hope, given our response, that the reviewer will reconsider their position. We thank the reviewer for this process, which helps us improve clarity in our work.
>
> ---
>
> **Q1:**  “Reproduce the given objective function when substituting Eqs. 5 and 6 into Eq. 4”
>
> **A1:** We very much appreciate the reviewer going through our proofs. The reason you could not re-deduce the objective is because of a type-O in our appendix: The appendix uses $1/\lambda$ instead of $\lambda$ multiplying the KL divergence. That is, in the appendix we have considered the objective
> $$ - \log p(g_{k+1} \mid m, \sigma^2) + \tfrac{1}{\lambda }\mathrm{KL}\left(\mathcal{N}(m_k, \sigma_k^2)\,\|\,\mathcal{N}(m, \sigma^2)\right) $$
>
>
>  This is the only reason your deduced objective does not match the one in the objective in the proof. We apologize for this type-O. In our revision, we will fix this, and take your advice and give all the explicit steps of the proof, starting by stating
>  “Substituting (5) and (6) into (4) gives the objective …”' This will result in a clearer proof.
>
> ---
>
> **Q2:** "I do not understand how the authors went from  $\beta \delta\_k + \beta(1-\beta)(m\_k-g\_{k+1})^2$ to $\beta EMA\{\beta}(m\_k - g\_{k+1})^2$?.. provide derivation, …  may increase my score."
>
> **A2:** Certainly, this derivation relies on applying the recursive definition of EMA given in Eq (1).   The short answer, is that the equivalence holds because the EMA as an operator is linear, and consequently $\beta EMA_{\beta}[(m_{k-1}-g_k)^2] =EMA_{\beta}[\beta(m_{k-1}-g_k)^2].$ In case it helps, we also give a more detailed proof below using induction.
>
> Our objective is to prove that, given $\delta_k$ defined recursively via:
>
> $\delta\_{k+1} =  \beta \delta\_k + \beta(1-\beta)(m\_k-g\_{k+1})^2 $
>
> it follows that  $\delta_k = \beta EMA_{\beta}[(m_{k-1}-g_k)^2]$, where by definition the EMA is also defined recursively via
>
> $ EMA_{\beta}[(m_{k}-g_{k+1})^2] = \beta EMA_{\beta}[(m_{k-1}-g_k)^2] +(1-\beta) (m\_k-g\_{k+1})^2.$
>
>
>
> **Base case**. First consider the base case $k=0,$ where we initialize $m_{-1}=m_0 = g_0,$ and thus
>
>  $\delta\_0 = 0 = \beta (m_{-1}-g_0)^2 = \beta EMA\_{\beta}[(m_{-1}-g_0)^2]$,
>
> since we also initialize EMA at the first element of the sequence. Thus the equivalence holds for $k=0.$
>
>
> **Induction hypothesis**. For $k>0$, assume we have that $\delta_k = \beta EMA_{\beta}[(m_{k-1}-g_k)^2],$ and let us now prove the equivalence for $k+1$ using this induction hypothesis. By the recursive definition of $\delta_{k+1}$ we have that
>
> $\delta_{k+1} = \beta \delta_k + \beta (1-\beta) (m_k - g_{k+1})^2  $
>
> $= \beta^2 EMA_{\beta}[(m_{k-1}-g_k)^2] + \beta (1-\beta) (m_k - g_{k+1})^2$
>
> $= \beta( \beta EMA_{\beta}[(m_{k-1}-g_k)^2] +  (1-\beta) (m_k - g_{k+1})^2)$
>
> $=  \beta EMA_{\beta}[(m_{k}-g_{k+1})^2]$
>
> Where in the second equality we used the induction hypothesis, and in the last equality we used the recursive definition of EMA.
>
>
> ---
>
> **Q3:**  interpretation of the $m_k^2$  factor in the denominator of Eq. 3?
>
> **A3:** The interpretation we give is that, because of the denominator which includes the $m_k^2$ term, we can re-write Adam (for $\beta_1=\beta_2$) as Eq (9), where it is clear that Adam is a mollified variant of Signum.
>
> ---
>
> **Q4:** There are several small typos in the maths
>
> **A4:** Thank you very much for these, and reading our paper and appendix so carefully. We corrected all of them in our latex file.
>
> ---
>
> **Q5:** Could we see a comparison of Adam and the update proposed in Eq.9?
>
> **A5:** The method in Eq. 9 is equivalent to Adam with $\beta_1 =\beta_2$ and $\epsilon=0$, which is the method we most experimented with.
>
> ---
>
> **Q6:** What regard their claim that Adam is an online estimate of the mean and variance of the gradient is novel compared to what is stated in the initial paper?
>
> **A5:** We apologize that we may not have understood your question here. When you ask “novel compared to what is stated in the initial paper”, here we assume you mean the origin Adam paper [1]. Please let us know if this is not the case, and we will gladly interact and answer.
>
> This viewpoint we give of Adam in Theorem 4.1 as a formal online estimator of mean and variance is entirely new. Despite years of research into Adam, to the best of our knowledge, we have never seen this viewpoint of Adam before. As compared to the original Adam paper [1], the authors introduce Adam as a “ ... the method computes individual adaptive learning rates for different parameters from estimates of first and second moments of the gradients”. That is, they refer to the $v_k$ buffer as a type of second moment estimator of the gradient. But there is no formal proof, or variational viewpoint that shows this. Compare this to our Theorem 4.1 that formally shows how Adam arise from finding the mean and variance that maximizes the likelihood of observing a gradient drawn from a Gaussian, subject to KL regularization term that encodes how the distribution over gradients changes gradually over iterations.
>
> The only formal proof in the original Adam paper [1] is Theorem 4.1, which attempts to establish the online regret of the iterates of Adam, and which is a notoriously incorrect proof (a counter example emerged in [2])
>
> To conclude, Let us also emphasize a crucial aspect of our theory: it is derived from the $\beta_1=\beta_2$ insight. Proposition 2 in the appendix shows that under no other choice we can get an explicit variance-like term in the reformulation. This is important, since the connection to SignSGD with momentum is only possible only if one can properly "split" the second order term $v$ into a squared mean and variance term. To the best of our knowledge, we are also the first to analyze this setting.
>
>
>
> [1] Adam: A Method for Stochastic Optimization, Diederik P. Kingma, Jimmy Ba, ICLR 2014
>
> [2] Sashank J. Reddi, Satyen Kale, and Sanjiv Kumar. "On the Convergence of Adam and Beyond."
> International Conference on Learning Representations, ICLR 2018.

---

> > ### Comment · Reviewer_rfMp · 2025-08-04
> > **Thank you for your answer**
> >
> > I thank the authors for their detailed and thoughtful answer, which has fully alleviated my concerns. I have updated my score accordingly.

---

### Official Review · Reviewer_9fwq · 2025-07-02

**Clarity:** 4
**Significance:** 3
**Originality:** 3
**Rating:** 5
**Confidence:** 3

**Summary:**

This is a very well-written submission offering a thorough analysis of the effectiveness of the Adam optimizer both experimentally and theoretically, arriving at a novel interpretation of the essential features of Adam, in turn leading to a set of simple, specific recommendations for best practices.

**Questions:**

Fig. 2, increase some of the fonts in the figure legends/labels? (Same for Fig. 4). Also, indicate more clearly that the top row is for Signum, and the bottom row is for Adam?

Equation right before L90, defining `clip`. If abbreviating to "Gclip" in the text, maybe define `gclip` or `Gclip` in the equation?

L111, what is a "landscape-based argument"?

L114, "less hyperparameters" --> "fewer hyperparameters"

L119, remove superfluous "4"?

L229, "Proposition 1": state that the proof is in the Appendix?

re: Theorem 4.1: state that the proof is in the Appendix?

The Conclusion seems to underplay the key theoretical findings, e.g. Proposition 1 and Theorem 4.1 (and the main other derivations in the Appendix). It seems the Conclusion could be expanded?

**Ethical Concerns:**

["NO or VERY MINOR ethics concerns only"]

**Limitations:**

Yes.

**Paper Formatting Concerns:**

None.

**Quality:**

4

**Strengths And Weaknesses:**

The main strengths of the work are: very high quality writing and overall excellent presentation, thorough and large-scale experimental grounding of the conceptual analysis, and very clear (at least in the foundation) theoretical analysis; elegant and simple concluding recommendations. The experimental evaluation should serve as a good reference for the behavior of Adam, Signum (and Sign) for the broader community. Also, AFAICT the work provides a good summary of relevant work in the community over the years.

The main weakness is what is IMO a slightly unclear, disconnected relationship between main body and Appendix.  The main body clearly depends on the Appendix material in several places, but this is not really mentioned in the main body. Conversely, there seems to be a fair bit of material in the Appendix that, though interesting and related, does not directly support anything in the main body. Ideally, there would be more of an organize, obvious relationship between main body and Appendix.

Other weaknesses are relatively minor: the figures are a bit cramped/unclear in places, and some parts of the text could be clarified too.

---

> ### Author Rebuttal · Authors · 2025-07-29
>
> We thank the reviewer for their work and their pleasant comments on our paper! We also thank them for their questions and suggestions, which will help further improve the quality of our work.
>
> We provide separate answers for your main points.
>
> ---
>
> **1) Disconnected Appendix**
>
> The reviewer is absolutely right, the main reason for this is that we did not want to mention precise places in the appendix due to the separate main paper and appendix deadlines (e.g., adding a new figure would compromise the ordering stated in the main paper). We will make sure everything in the appendix is linked semantically to the main paper and properly connected. Another reason for this disconnect was the tight space in the main paper, insufficient to properly comment all our side results. This can be though of course done in the potential camera-ready.
>
> ---
>
> **2) Improved Figures and Conclusion**
>
> We are committed to delivering the best possible graphical results, and therefore will devote efforts to addressing all reviewers' comments, such as those regarding Figure 4. We will also expand our conclusion section to address the theoretical aspects of the paper better.
>
> ---
>
> **3) Other points**
>
> 1. We will define Gclip in the equation. This helps clarity, thanks!
> 2. With "landscape-based arguments" we mean arguments that attribute the Adam-SGD gap not to noise in the gradient but on the Hessian structure of Transformers. An example is the paper "Why Transformers Need Adam: A Hessian Perspective" by Zhang et al. We will specify in our updated version.
> 3. Typos and other suggestions: thanks so much, we already updated our LaTeX file with your comments!

---

### Official Review · Reviewer_89qA · 2025-07-03

**Clarity:** 3
**Significance:** 3
**Originality:** 2
**Rating:** 5
**Confidence:** 3

**Summary:**

This paper presents an extensive empirical study on why Adam works so well for training Transformer models and reports two key observations:
* Signum closes most of the perplexity gap between SGD and Adam but still suffers from a slowdown, hinting that Adam contains additional “secret sauce.”
* The larger the chosen $\beta_1$, the larger the optimal $\beta_2$. $\beta_1 = \beta_2$ is near-optimal.

The authors then reinterpret Adam with $\beta_1 = \beta_2$ as a mollified version of Signum whose mollification factor is the local gradient noise-to-signal ratio. They argue this adaptive mollification is Adam’s hidden advantage and confirm the claim in a controlled setting.

**Questions:**

1. Zhao et al. report that the performance of Signum matches Adam well. What do you think might be the reason of the discrepancy?
2. In a recent work, Srećković et al. argue that batch size plays an important role in the gap between Adam and SGD. Do you think Signum can match Adam if the batch size is better tuned? How do you think batch size will affect the "secret sauce" of Adam?

Reference

[1] Zhao, Rosie, et al. "Deconstructing What Makes a Good Optimizer for Language Models." OPT 2024: Optimization for Machine Learning.

[2] Srećković, Teodora, Jonas Geiping, and Antonio Orvieto. "Is your batch size the problem? Revisiting the Adam-SGD gap in language modeling." arXiv:2506.12543.

**Ethical Concerns:**

["NO or VERY MINOR ethics concerns only"]

**Limitations:**

Yes.

**Paper Formatting Concerns:**

It is unlikely to be a severe problem, but the caption of the figures are in italics.

**Quality:**

3

**Strengths And Weaknesses:**

Strengths
* The authors conducted extensive grid search on hyperparameters and claim to open the experiment data to the public. This not only substantiates the findings of the paper, but also facilitates future research on this topic.
* The results challenge the common $\beta_1=0.9$, $\beta_2=0.95$ Adam training recipe for LLMs and suggest that $\beta_2=0.9$ might be the better default, potentially saving compute and simplifying future hyper-parameter tuning for Adam-like optimizers.

Weaknesses
* As shown in Fig. 2-4, $\beta_1 = \beta_2$ generally performs well but is not always the optimal choice, implying additional factors affecting the training dynamics of Adam.
* This paper focus on Chinchilla-optimal settings, whereas many modern LLMs are trained on far more data. It is unclear whether the reported trends still hold in such regime.

---

> ### Author Rebuttal · Authors · 2025-07-28
>
> We thank the reviewer for their reading and appreciation of our work. It truly means a lot to us to hear that you also believe our empirical evidence to be extensive and of practical interest — especially for future research.
>
> We also like to answer in detail your questions, as we believe those are very interesting: we will highlight these points in our revised manuscript and also include the new results we present next.
>
> ---
>
> **1) Optimality**
>
> While $\beta_1=\beta_2$ performs in our experiments better than the usual choice $0.9, 0.95$ we do not claim that $\beta_1=\beta_2$ is always the best choice -- but instead that
>
> a. it is quite convenient for tuning, while leading to near-optimal results.
>
> b. brings about theoretical insights.
>
> ---
>
> **2) Training on more tokens.**
>
> This is a setting that definitely helps deliver our point, and we thank the reviewer for bringing this up. We were able to run a quite comprehensive ablation. The results below are in the setting of Figure 2, but we train for 6.4B tokens instead of 3.2B. We performed a total of 168 full training runs, each taking approximately 10 hours on a A100 with 80GB. For each choice of $\beta_1,\beta_2\in [0.9, 0.95, 0.975, 0.9875]\times[0.8 ,0.9, 0.95, 0.975, 0.9875, 0.99375]$ we report the best learning rate in the grid $[0.005, 0.001, 0.002, 0.004, 0.008, 0.016, 0.032]$. We highlight the choice $\beta_1=\beta_2$ below. We report validation perplexity.
>
> | β₁  / β₂  | 0.8    | 0.9    | 0.95   | 0.975  | 0.9875 | 0.99375 |
> |---------------|--------|--------|--------|--------|--------|---------|
> | **0.9**           | 20.15  | ***20.00***  | 20.00  | 19.94  | 20.14  | 20.8    |
> | **0.95**          | 20.14  | 19.93  | ***19.87***  | 19.93  | 19.88  | 20.24   |
> | **0.975**         | 262.12 | 20.01  | 19.80  | ***19.73***  | 19.72  | 19.74   |
> | **0.9875**        | 5797      | 189    | 20.40     | 19.88  | ***19.81***  |  19.87 |
>
> As the reviewer can notice, $\beta_1=\beta_2$ is always near-optimal (gap of 0.01 ppl), with a slight imperfection at $\beta_1=0.9$. Note that this imperfection is also quite apparent in Figure 2. This does not quite matter because $\beta_1=0.9$ leads anyways to suboptimal results. Notice how, as reported in Figure 3 in the paper, we notice a strong correlation between $\beta_1$ and $\beta_2$: the higher $\beta_1$, the higher $\beta_2$ has to be for optimality.
>
> Note that crucially, if one only tries values on the diagonal $\beta_1=\beta_2$, one gets to a perplexity which is only suboptimal by 0.01 ppl. This type of optimality also holds in Fig. 2 and 4.
>
> ---
>
> **3) Question: Signum vs Adam + Batch Size effect**
>
> This is a crucial point, which we again thank the reviewer for bringing up, this is worth a paragraph in our potential camera-ready. We believe our results are in *perfect agreement* with the 2 papers you mention. We would, however, like to note that we have already partially addressed this point in our paper (lines 196 in the main text and 491 in the appendix).
>
> Let us expand: we first summarize what the two papers you mention say and then compare them with our claims.
>
> ***Summary***: Zhao et al. report that Signum and Adam perform quite similarly. Srećković et al. show that the gap between Adam and SGD shrinks to almost zero at very low batch-sizes. Srećković et al. was published after the Neurips deadline, so that is why we could not compare with this work.
>
> ***Comparison***: The finding of  Srećković et al. was additionally confirmed by [A], which goes one step further and test a few optimizers other than Adam and SGD. In particular, they compare Adam with Adafactor — which (similarly to Signum) can be seen as a simplification of Adam. Note how in Figure 1(a) in [A] they show how the gap between Adam and Adafactor shrinks as the batch size decreases. We also notice a similar fact and already commented in line 196 of our submission, pointing to our Figure 12 in the appendix: at a shorter sequence length of 512 — i.e. a smaller number of tokens/iteration — the gap between Signum and Adam shrinks. We did this experiment to compare indeed with Zhao et al., that uses a fixed sequence length of 512. Hence, we believe our results are in perfect agreement with the 2 papers you mention.
>
> There is a final important point to stress regarding shrinkage of the gap at low batch sizes: this setting is not practical for parallel computing, where batch sizes are often a standard. Our result in Figure 1 illustrates the gap between Signum and Adam in a practical scenario, with commonly used values of batchsize $\times$ sequence length.
>
>
> [A] Marek, Martin, et al. "Small Batch Size Training for Language Models: When Vanilla SGD Works, and Why Gradient Accumulation Is Wasteful." arXiv preprint arXiv:2507.07101 (2025).

---

> ### Comment · Reviewer_89qA · 2025-08-01
>
> Thank you for the clarification and additional experiments! They have helped me gain a better understanding of the paper.
>
> I have two remaining concerns. I understand they may not be easily addressed in a short time, and this doesn't affect my view of the paper as a valuable contribution  (I'm not sure if I will raise the score further, as I believe a strong accept should be reserved for truly exceptional cases.)
> 1. The theoretical and empirical results on $\beta_1=\beta_2$ are indeed important, as they offer a new perspective and simplify hyperparameter tuning a lot. Still, the fact that $\beta_1=\beta_2$ is *near*-optimal rather than truly optimal suggests that there are other components in the "secret sauce" of AdamW beyond what is identified in this paper.
> 2. As pointed out by the authors, this paper leads to similar conclusions to prior work that AdamW works better than Signum when batch size is large. However, according to eq. (9), when $\sigma$ decreases (which should occur when batch size increases), AdamW should be more similar to Signum, and this doesn't seem to align with the empirical results.

---

### Comment · Area_Chair_ZAAp · 2025-08-04
**Gentle Reminder: Reviewer Response to Rebuttal Needed**

Dear reviewers,

Thank you for your valuable time and expertise in reviewing. The author rebuttal phase is nearing its close, and we kindly request your prompt attention to ensure a thorough discussion.

**The discussion period ends in less than 3 days (on Aug. 6, 11:59pm AOE )**. To maintain the review timeline, we ask that you:

1. Review the authors’ rebuttal,

2. Engage in any ongoing discussion with fellow reviewers/authors (if applicable),

3. Finalize your assessment.


If you have already completed this step, please disregard this reminder—we sincerely appreciate your efforts.
Your timely input is crucial to the integrity of our review process. Thank you for your collaboration!

Best regards,

AC

---

### Note · Authors · 2025-08-12

Dear Reviewers,

We thank you so much for this interaction, for your suggestions, and the very valuable feedback on our work.

This paper stemmed from our scientific curiosity, and our only purpose is to produce valuable insights for the community.

 With your help, our manuscript has been further improved: in particular,

- We performed additional experiments as suggested by 89qA, training models beyond the Chinchilla-optimal budget. The results are aligned with our claims, and be believe this was a very crucial experiment to run.

- Following the directions by rfMp we corrected a typo in the proof and made some points clearer. We are happy our reply could solve their concern.

We are hopeful our work, together with the github repo we will publish, can serve as a resource for researchers looking to understand adaptive methods. These are exciting times, and we are happy to have contributed with a thorough optimizer ablation at this important stage for LM optimization research.

---

### Decision · Program_Chairs · 2025-09-17

**Decision:**

Accept (oral)

**Comment:**

This paper presents a thorough and compelling investigation into the Adam optimizer, a cornerstone of modern deep learning. Through an extensive empirical study encompassing over 1,300 language model trainings, the authors arrive at a significant and practical finding: constraining Adam's momentum parameters to be equal ($\beta_1 = \beta_2$) yields robust, near-optimal performance. This discovery is substantiated by a novel theoretical interpretation, framing Adam under this configuration as a natural online algorithm for estimating gradient statistics, derived through a mean-field variational inference perspective.

The manuscript is well-written, clearly organized, and makes several key contributions to the community:

- It provides a definitive empirical analysis, demonstrating that while Signum narrows the performance gap with SGD, Adam's adaptive trust region—governed by the signal-to-noise ratio of the gradients—consistently provides a measurable advantage.

- It offers a valuable practical guideline, showing that the simple constraint $\beta_1 = \beta_2$ is highly effective, thereby significantly reducing the hyperparameter search space for practitioners.

- It delivers a profound theoretical insight, reformulating Adam's update rule through a clean variational inference framework that elucidates its underlying mechanics.

The authors have satisfactorily addressed all concerns raised by the reviewers during the discussion period. It is noted that the related work section is comprehensive, though it miss a relevant work (https://openreview.net/forum?id=TBJCtWTvXJ), which provides a complementary theoretical justification for $\beta_1 = \beta_2$ from the perspective of mitigating loss spikes.

In summary, this work is grounded in extensive experimentation and deep theoretical analysis. Its insightful conclusions regarding both the performance and interpretation of Adam are timely and will be of substantial value to the  community. For these reasons, I strongly recommend acceptance.